# Unsupervised Learning of Full-Waveform Inversion: Connecting CNN and Partial Differential Equation in a Loop

**Peng Jin**[1,2,*], **Xitong Zhang**[1,3,*], **Yinpeng Chen**[4], **Sharon Xiaolei Huang**[2],
**Zicheng Liu**[4] **and Youzuo Lin**[1]

[1]Earth and Environmental Sciences Division, Los Alamos National Laboratory
[2]College of Information Sciences and Technology, The Pennsylvania State University
[3]Department of Computational Mathematics, Science and Engineering, Michigan State University
[4]Microsoft Research

{pqj5125,suh972}@psu.edu,zhangxit@msu.edu,{yiche,zliu}@microsoft.com,
ylin@lanl.gov

## Abstract

This paper investigates unsupervised learning of Full-Waveform Inversion (FWI), which has been widely used in geophysics to estimate subsurface velocity maps from seismic data. This problem is mathematically formulated by a second order partial differential equation (PDE), but is hard to solve. Moreover, acquiring velocity map is extremely expensive, making it impractical to scale up a supervised approach to train the mapping from seismic data to velocity maps with convolutional neural networks (CNN).We address these difficulties by *integrating PDE and CNN in a loop*, thus shifting the paradigm to unsupervised learning that only requires seismic data. In particular, we use finite difference to approximate the forward modeling of PDE as a differentiable operator (from velocity map to seismic data) and model its inversion by CNN (from seismic data to velocity map). Hence, we transform the supervised inversion task into an unsupervised seismic data reconstruction task. We also introduce a new large-scale dataset *OpenFWI*, to establish a more challenging benchmark for the community. Experiment results show that our model (using seismic data alone) yields comparable accuracy to the supervised counterpart (using both seismic data and velocity map). Furthermore, it outperforms the supervised model when involving more seismic data.

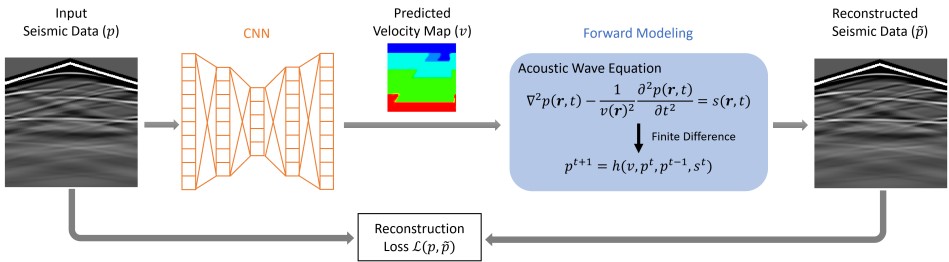

Figure 1: Schematic illustration of our proposed method, which comprises a CNN to learn an inverse mapping and a differentiable operator to approximate the forward modeling of PDE.

## 1 Introduction

Geophysical properties (such as velocity, impedance, and density) play an important role in various subsurface applications including subsurface energy exploration, carbon capture and sequestration,

---

*Equal contribution.
Dataset is available at https://openfwi-lanl.github.io.

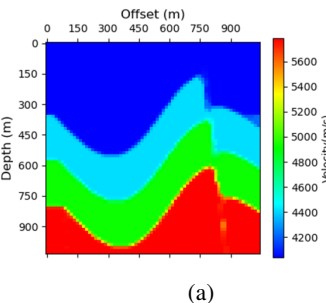 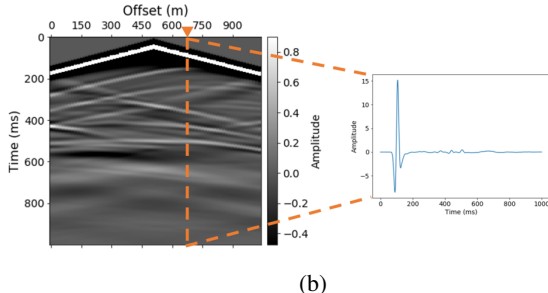

(a)                                      (b)

Figure 2: An example of (a) a velocity map and (b) seismic measurements (named shot gather in geophysics) and the 1D time-series signal recorded by a receiver.

estimating pathways of subsurface contaminant transport, and earthquake early warning systems to provide critical alerts. These properties can be obtained via seismic surveys, i.e., receiving reflected/refracted seismic waves generated by a controlled source. This paper focuses on reconstructing subsurface velocity maps from seismic measurements. Mathematically, the velocity map and seismic measurements are correlated through an acoustic-wave equation (a second-order partial differential equation) as follows:

$$\nabla^2 p(\boldsymbol{r}, t) - \frac{1}{v(\boldsymbol{r})^2} \frac{\partial^2 p(\boldsymbol{r}, t)}{\partial t^2} = s(\boldsymbol{r}, t) \,, \tag{1}$$

where $p(\boldsymbol{r}, t)$ denotes the pressure wavefield at spatial location $\boldsymbol{r}$ and time $t$, $v(\boldsymbol{r})$ represents the velocity map, and $s(\boldsymbol{r}, t)$ is the source term. Full-Waveform Inversion (FWI) is a methodology that determines high-resolution velocity maps $v(\boldsymbol{r})$ of subsurface via matching synthetic seismic waveforms to raw recorded seismic data $p(\tilde{\boldsymbol{r}}, t)$, where $\tilde{\boldsymbol{r}}$ represents the locations of seismic receivers.

A velocity map describes the wave propagation speed in the subsurface region of interest. An example in 2D scenario is shown in Figure 2a. Particularly, the x-axis represents the horizontal offset of a region, and the y-axis stands for the depth. The regions with the same geologic information (velocity) are called a layer in velocity maps. In a sample of seismic measurements (termed a shot gather in geophysics) as depicted in Figure 2b, each grid in the x-axis represents a receiver, and the value in the y-axis is a 1D time-series signal recorded by each receiver.

Existing approaches solve FWI in two directions: *physics-driven* and *data-driven*. Physics-driven approaches rely on the forward modeling of Equation 1, which simulates seismic data from velocity map by finite difference. They optimize velocity map per seismic sample, by iteratively updating velocity map from an initial guess such that simulated seismic data (after forward modeling) is close to the input seismic measurements. However, these methods are slow and difficult to scale up as the iterative optimization is required per input sample. Data-driven approaches consider FWI problem as an image-to-image translation task and apply convolution neural networks (CNN) to learn the mapping from seismic data to velocity maps (Wu & Lin, 2019). The limitation of these methods is that they require paired seismic data and velocity maps to train the network. Such ground truth velocity maps are hardly accessible in real-world scenario because generating them is extremely time-consuming even for domain experts.

In this work, we leverage advantages of both directions (physics + data driven) and shift the paradigm to unsupervised learning of FWI by connecting forward modeling and CNN in a loop. Specifically, as shown in Figure 1, a CNN is trained to predict a velocity map from seismic data, which is followed by forward modeling to reconstruct seismic data. The loop is closed by applying reconstruction loss on seismic data to train the CNN. Due to the differentiable forward modeling, the whole loop can be trained end-to-end. Note that the CNN is trained in an unsupervised manner, as the ground truth of velocity map is *not* needed. We name our unsupervised approach as UPFWI (Unsupervised Physics-informed Full-Waveform Inversion).

Additionally, we find that perceptual loss (Johnson et al., 2016) is crucial to improve the overall quality of predicted velocity maps due to its superior capability in preserving the coherence of the reconstructed waveforms comparing with other losses like Mean Squared Error (MSE) and Mean Absolute Error (MAE).

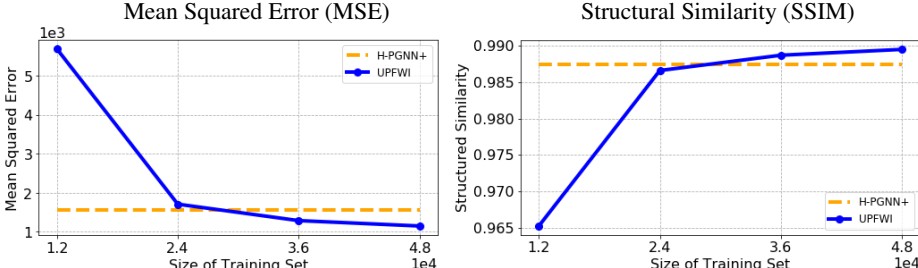

Figure 3: **Unsupervised UPFWI (ours) vs. Supervised H-PGNN+ (Sun et al., 2021)**. Our method achieves better performance, e.g. lower Mean Squared Error (MSE) and higher Structural Similarity (SSIM), when involving more unlabeled data (>24k).

To encourage fair comparison on a large dataset with more complicate geological structures, we introduce a new synthetic dataset named OpenFWI, which contains 60,000 labeled data (velocity map and seismic data pairs) and 48,000 unlabeled data (seismic data alone). 30,000 of those velocity maps contain curved layers that are more challenge for inversion. We also add geological faults with various shift distances and tilting angles to all velocity maps.

We evaluate our method on this dataset. Experimental results show that for velocity maps with flat layers, our UPFWI trained with 48,000 unlabeled data achieves 1146.09 in MSE, which is 26.77% smaller than that of the supervised baseline H-PGNN+ (Sun et al., 2021), and 0.9895 in Structured Similarity (SSIM), which is 0.0021 higher; for velocity maps with curved layers, our UPFWI achieves 3639.96 in MSE, which is 28.30% smaller, and 0.9756 in SSIM, which is 0.0057 higher.

Our contribution is summarized as follows:

- We propose to solve FWI in an unsupervised manner by connecting CNN and forward modeling in a loop, enabling end-to-end learning from seismic data alone.
- We find that perceptual loss is helpful to boost the performance comparable to the supervised counterpart.
- We introduce a large-scale dataset as benchmark to encourage further research on FWI.

## 2 PRELIMINARIES OF FULL-WAVEFORM INVERSION (FWI)

The goal of FWI in geophysics is to invert for a velocity map $v \in \mathbb{R}^{W \times H}$ from seismic measurements $p \in \mathbb{R}^{S \times T \times R}$, where $W$ and $H$ denote the horizontal and vertical dimensions of the velocity map, $S$ is the number of sources used to generate waves during data acquisition process, $T$ denotes the number of samples in the wavefields recorded by each receiver, and $R$ represents the total number of receivers.

In conventional physics-driven methods, forward modeling is commonly referred to the process of simulating seismic data $\tilde{p}$ from given estimated velocity maps $\hat{v}$. For simplicity, the forward acoustic-wave operator $f$ can be expressed as

$$\tilde{p} = f(\hat{v}) . \tag{2}$$

Given this forward operator $f$, the physics-driven FWI can be posed as a minimization problem (Virieux & Operto, 2009)

$$E(\hat{v}) = \min_{\hat{v}} \left\{ ||p - f(\hat{v})||_2^2 + \lambda R(\hat{v}) \right\} , \tag{3}$$

where $||p - f(\hat{v})||_2^2$ is the the $\ell_2$ distance between true seismic measurements $p$ and the corresponding simulated data $f(\hat{v})$, $\lambda$ is a regularization parameter and $R(\hat{v})$ is the regularization term which is often the $\ell_2$ or $\ell_1$ norm of $\hat{v}$. This requires optimization per sample, which is slow as the optimization involves multiple iterations from an initial guess.

Data-driven methods leverage CNNs to directly learn the inverse mapping as (Adler et al., 2021)

$$\hat{v} = g_\theta(p) \approx f^{-1}(p) , \tag{4}$$

where $g_\theta(\cdot)$ is the approximated inverse operator of $f(\cdot)$ parameterized by $\theta$. In practice, $g_\theta$ is usually implemented as a CNN (Adler et al., 2021; Wu & Lin, 2019). This requires paired seismic data and velocity maps for supervised learning. However, the acquisition of large volume of velocity maps in field applications can be extremely challenging and computationally prohibitive.

## 3 METHOD

In this section, we present our Unsupervised Physics-informed solution (named UPFWI), which connects CNN and forward modeling in a loop. It addresses limitations of both physics-driven and data-driven approaches, as it requires neither optimization at inference (per sample), nor velocity maps as supervision.

### 3.1 UPFWI: CONNECTING CNN AND FORWARD MODELING

As depicted in Figure 1, our UPFWI connects a CNN $g_\theta$ and a differentiable forward operator $f$ to form a loop. In particular, the CNN takes seismic measurements $p$ as input and generates the corresponding velocity map $\hat{v}$. We then apply forward acoustic-wave operator $f$ (see Equation 2) on the estimated velocity map $\hat{v}$ to reconstruct the seismic data $\tilde{p}$. Typically, the forward modeling employs finite difference (FD) to discretize the wave equation (Equation 1). The details of forward modeling will be discussed Section 3.3. The loop is closed by the reconstruction loss between input seismic data $p$ and reconstructed seismic data $\tilde{p} = f(g_\theta(p))$. Notice that the ground truth of the velocity map $v$ is not involved, and the training process is *unsupervised*. Since the forward operator is differentiable, the reconstruction loss can be backpropagated (via gradient descent) to update the parameters $\theta$ in the CNN.

### 3.2 CNN NETWORK ARCHITECTURE

We use an encoder-decoder structured CNN (similar to Wu & Lin (2019) and Zhang & Lin (2020)) to model the mapping from seismic data $p \in \mathbb{R}^{S \times T \times R}$ to velocity map $v \in \mathbb{R}^{W \times H}$. The encoder compresses the seismic input and then transforms the latent vector to build the velocity estimation through a decoder. See the implementation details in Appendix A.1.

### 3.3 DIFFERENTIABLE FORWARD MODELING

We apply the standard finite difference (FD) in the space domain and time domain to discretize the original wave equation. Specifically, the second-order central finite difference in time domain ($\frac{\partial^2 p(\boldsymbol{r}, t)}{\partial t^2}$ in Equation 1) is approximated as follows:

$$\frac{\partial^2 p(\boldsymbol{r}, t)}{\partial t^2} \approx \frac{1}{(\Delta t)^2}(p_{\boldsymbol{r}}^{t+1} - 2p_{\boldsymbol{r}}^t + p_{\boldsymbol{r}}^{t-1}) + O[(\Delta t)^2] \, , \tag{5}$$

where $p_{\boldsymbol{r}}^t$ denotes the pressure wavefields at timestep $t$, and $p_{\boldsymbol{r}}^{t+1}$ and $p_{\boldsymbol{r}}^{t-1}$ are the wavefields at $t + \Delta t$ and $t - \Delta t$, respectively. The Laplacian of $p(\boldsymbol{r}, t)$ can be estimated in the similar way on the space domain (see Appendix A.2). Therefore, the wave equation can then be written as

$$p_{\boldsymbol{r}}^{t+1} = (2 - v^2 \nabla^2)p_{\boldsymbol{r}}^t - p_{\boldsymbol{r}}^{t-1} - v^2(\Delta t)^2 s_{\boldsymbol{r}}^t \, , \tag{6}$$

where $\nabla^2$ here denotes the discrete Laplace operator.

The initial wavefield at the initial timestep is set to zero (i.e. $p_{\boldsymbol{r}}^0 = 0$). Thus, the gradient of loss $\mathcal{L}$ with respect to estimated velocity at spatial location $\boldsymbol{r}$ can be computed using the chain rule as

$$\frac{\partial \mathcal{L}}{\partial v(\boldsymbol{r})} = \sum_{t=0}^{T} \left[ \frac{\partial \mathcal{L}}{\partial p(\boldsymbol{r}, t)} \right] \frac{\partial p(\boldsymbol{r}, t)}{\partial v(\boldsymbol{r})} \, , \tag{7}$$

where $T$ indicates the total number of timesteps.

### 3.4 Loss Function

The reconstruction loss of our UPFWI includes a pixel-wise loss and a perceptual loss as follows:

$$\mathcal{L}(\boldsymbol{p}, \tilde{\boldsymbol{p}}) = \mathcal{L}_{pixel}(\boldsymbol{p}, \tilde{\boldsymbol{p}}) + \mathcal{L}_{perceptual}(\boldsymbol{p}, \tilde{\boldsymbol{p}}), \tag{8}$$

where $\boldsymbol{p}$ and $\tilde{\boldsymbol{p}}$ are input and reconstructed seismic data, respectively. The pixel-wise loss $\mathcal{L}_{pixel}$ combines $\ell_1$ and $\ell_2$ distance as:

$$\mathcal{L}_{pixel}(\boldsymbol{p}, \tilde{\boldsymbol{p}}) = \lambda_1 \ell_1(\boldsymbol{p}, \tilde{\boldsymbol{p}}) + \lambda_2 \ell_2(\boldsymbol{p}, \tilde{\boldsymbol{p}}), \tag{9}$$

where $\lambda_1$ and $\lambda_2$ are two hyper-parameters to control the relative importance. For the perceptual loss $\mathcal{L}_{perceptual}$, we extract features from `conv5` in a VGG-16 network (Simonyan & Zisserman, 2015) pretrained on ImageNet (Krizhevsky et al., 2012) and combine the $\ell_1$ and $\ell_2$ distance as:

$$\mathcal{L}_{perceptual}(\boldsymbol{p}, \tilde{\boldsymbol{p}}) = \lambda_3 \ell_1(\phi(\boldsymbol{p}), \phi(\tilde{\boldsymbol{p}})) + \lambda_4 \ell_2(\phi(\boldsymbol{p}), \phi(\tilde{\boldsymbol{p}})), \tag{10}$$

where $\phi(\cdot)$ represents the output of `conv5` in the VGG-16 network, and $\lambda_3$ and $\lambda_4$ are two hyper-parameters. Compared to the pixel-wise loss, the perceptual loss is better to capture the region-wise structure, which reflects the waveform coherence. This is crucial to boost the overall accuracy of velocity maps (e.g. the quantitative velocity values and the structural information).

## 4 OpenFWI Dataset

We introduce a new large-scale geophysics FWI dataset OpenFWI, which consists of 108K seismic data for two types of velocity maps: one with flat layers (named FlatFault) and the other one with curved layers (named CurvedFault). Each type has 54K seismic data, including 30K with paired velocity maps (labeled) and 24K unlabeled. The 30K labeled pairs are splitted as 24K/3K/3K for training, validation and testing respectively. Samples are shown in Appendix A.3.

The shape of curves in our dataset follows a sine function. Velocity maps in CurvedFault are designed to validate the effectiveness of FWI methods on curved topography. Compared to the maps with flat layers, curved velocity maps yield much more irregular geological structures, making inversion more challenging. Both FlatFault and CurvedFault contain 30,000 samples with 2 to 4 layers and their corresponding seismic data. Each velocity map has dimensions of 70×70, and the grid size is 15 meter in both directions. The layer thickness ranges from 15 grids to 35 grids, and the velocity in each layer is randomly sampled from a uniform distribution between 3,000 meter/second and 6,000 meter/second. The velocity is designed to increase with depth to be more physically realistic. We also add geological faults to every velocity map. The faults shift from 10 grids to 20 grids, and the tilting angle ranges from -123° to 123°.

To synthesize seismic data, five sources are evenly placed on surface with a 255-meter spacing, and seismic traces are recorded by 70 receivers at each grid with an interval of 15 meter. The source is a Ricker wavelet with a central frequency of 25 Hz (Wang, 2015). Each receiver records time-series data for 1 second, and we use a 1 millisecond sample rate to generate 1,000 timesteps. Therefore, the dimensions of seismic data become 5×1000×70. Compared to existing datasets (Yang & Ma, 2019; Moseley et al., 2020), OpenFWI is significantly larger. It includes more complicated and physically realistic velocity maps. We hope it establishes a more challenging benchmark for the community.

## 5 Experiments

In this section, we present experimental results of our proposed UPFWI evaluated on the OpenFWI.

### 5.1 Implementation Details

**Training Details:** The input seismic data are normalized to range $[-1, 1]$. We employ AdamW (Loshchilov & Hutter, 2018) optimizer with momentum parameters $\beta_1 = 0.9$, $\beta_2 = 0.999$ and a weight decay of $1 \times 10^{-4}$ to update all parameters of the network. The initial learning rate is set to $3.2 \times 10^{-4}$, and we reduce the learning rate by a factor of 10 when validation loss reaches a plateau. The minimum learning rate is set to $3.2 \times 10^{-6}$. The size of a mini-batch is set to 128. All

| Supervision | Method | FlatFault | | | CurvedFault | | |
|---|---|---|---|---|---|---|---|
| | | MAE ↓ | MSE ↓ | SSIM ↑ | MAE ↓ | MSE ↓ | SSIM ↑ |
| Supervised | InversionNet | 15.83 | 2156.00 | 0.9832 | 23.77 | 5285.38 | 0.9681 |
| | VelocityGAN | 16.15 | 1770.31 | 0.9857 | 25.83 | 5076.79 | 0.9699 |
| | H-PGNN+ (our implementation) | **12.91** | 1565.02 | 0.9874 | 24.19 | 5139.60 | 0.9685 |
| Unsupervised | **UPFWI-24K** (ours) | 16.27 | 1705.35 | 0.9866 | 29.59 | 5712.25 | 0.9652 |
| | **UPFWI-48K** (ours) | 14.60 | **1146.09** | **0.9895** | **23.56** | **3639.96** | **0.9756** |

Table 1: **Quantitative results evaluated on OpenFWI** in terms of MAE, MSE and SSIM. Our UPFWI yields comparable inversion accuracy comparing to supervised baselines. For H-PGNN+, we use our network architecture to replace the original one reported in their paper, and an additional perceptual loss between seismic data is added during training.

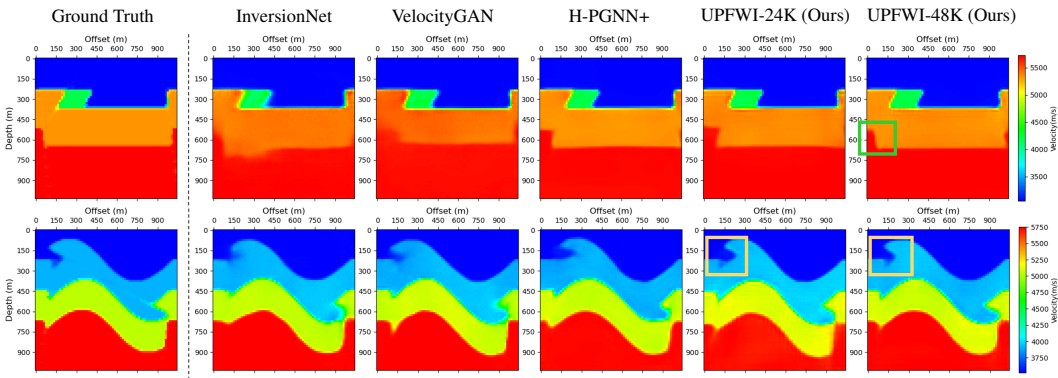

Figure 4: **Comparison of different methods on inverted velocity maps of FlatFault (top) and CurvedFault (bottom).** For FlatFault, our UPFWI-48K reveals more accurate details at layer boundaries and the slope of the fault in deep region. For CurvedFault, our UPFWI reconstructs the geological anomalies on the surface that best match the ground truth.

trade-off hyper-parameters $\lambda$ in our loss function are set to 1. We implement our models in Pytorch and train them on 8 NVIDIA Tesla V100 GPUs. All models are randomly initialized.

**Evaluation Metrics:** We consider three metrics for evaluating the velocity maps inverted by our method: MAE, MSE and SSIM. Both MAE and MSE have been employed in existing methods (Wu & Lin, 2019; Zhang & Lin, 2020) to measure pixel-wise errors. Considering the layered-structured velocity maps contain highly structured information, degradation or distortion can be easily perceived by a human. To better align with human vision, we employ SSIM to measure perceptual similarity. Note that for MAE and MSE calculation, we denormalize velocity maps to their original scale while we keep them in normalized scale [-1, 1] for SSIM according to the algorithm.

**Comparison:** We compare our method with three state-of-the-art algorithms: two pure data-driven methods, i.e., InversionNet (Wu & Lin, 2019) and VelocityGAN (Zhang & Lin, 2020), and a physics-informed method H-PGNN (Sun et al., 2021). We follow the implementation described in these papers and search for the best hyper-parameters for OpenFWI dataset. Note that we improve H-PGNN by replacing the network architecture with the CNN in our UPFWI and adding perceptual loss, resulting in a significant boosted performance. We refer our implementation as H-PGNN+, which is a strong supervised baseline. Our method has two variants (UPFWI-24K and UPFWI-48K), using 24K and 48K unlabeled seismic data respectively.

## 5.2  MAIN RESULTS

**Results on FlatFault:** Table 1 shows the results of different methods on FlatFault. Compared to InversionNet and VelocityGAN, our UPFWI-24K performs better in MSE and SSIM, but is slightly worse in MAE score. Compared to H-PGNN+, there is a gap between our UPFWI-24K and H-PGNN+ when trained with the same amount of data. However, after we double the size of unlabeled data (from 24K to 48K), a significant improvement is observed in our UPFWI-48K for all three metrics, and it outperforms all three supervised baselines in MSE and SSIM. This demonstrates the potential of our UPFWI for achieving higher performance with more unlabeled data involved.

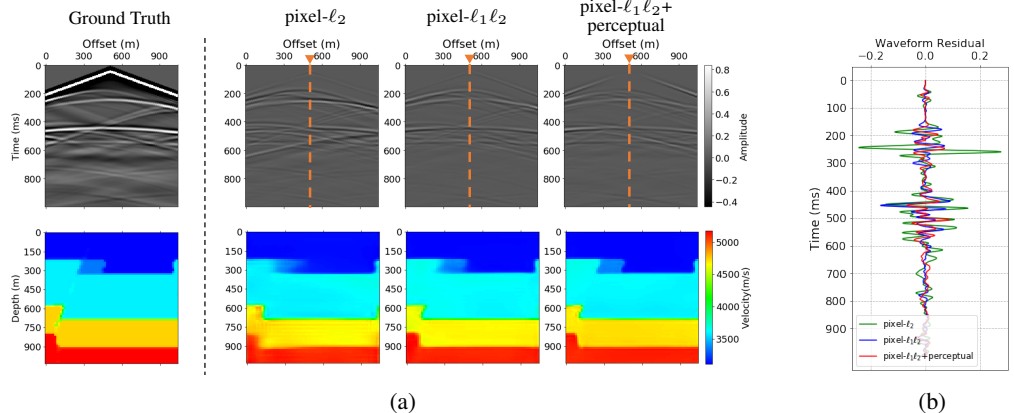

(a)                                                                   (b)

Figure 5: **Comparison of UPFWI with different loss functions** on (a) waveform residual and their corresponding inversion results (ground truth provided in the first column), and (b) single trace residuals recorded by the receiver at 525 m offset. Our UPFWI trained with pixel-wise loss ($\ell_1+\ell_2$ distance) and perceptual loss yields the most accurate results. Best viewed in color.

The velocity maps inverted by different methods are shown in the top row of Figure 4. Consistent with our quantitative analysis, more accurate details are observed in the velocity maps generated by UPFWI-48K. For instance, we find in the visualization results that both InversionNet and VelocityGAN generate blurry results in deep region, while H-PGNN+, UPFWI-24K and UPFWI-48K yield much clearer boundaries. We attribute this finding as the impact of seismic loss. We further observe that the slope of the fault in deep region is different from that in the shallow region, yet only UPFWI-48K replicates this result as highlighted by the green square.

**Results on CurvedFault:** Table 1 shows the results of CurvedFault. Performance degradation is observed for all models, due to the more complicated geological structures in CurvedFault. Although our UPFWI-24K underperforms the three supervised baselines, our UPFWI-48K significantly boosts the performance, outperforming all supervised methods in terms of all three metrics. This demonstrates the power of unsupervised learning in our UPFWI that greatly benefits from more unlabeled data when dealing with more complicated curved structure.

The bottom row of Figure 4 shows the visualized velocity maps in CurvedFault obtained using different methods. Similar to the observation in FlatFault, our UPFWI-48K yields more accurate details compared to the results of supervised methods. For instance, only our UPFWI-24K and UPFWI-48K precisely reconstruct the fault beneath the curve around the top-left corner as highlighted by the yellow square. Although some artifacts are observed in the results of UPFWI-24K around the layer boundary in deep region, they are eliminated in the results of UPFWI-48K. More visualization results are shown in Appendix A.3.

### 5.3 ABLATION STUDY

**Loss Terms:** We study the contribution of each loss term in our loss function: (a) pixel-wise $\ell_2$ distance (MSE), (b) pixel-wise $\ell_1$ distance (MAE), and (c) perceptual loss. All experiments are conducted on FlatFault using 24,000 unlabeled data.

| Loss | | | Velocity Error | | | Seismic Error | | |
|---|---|---|---|---|---|---|---|---|
| pixel-$\ell_2$ | pixel-$\ell_1$ | perceptual | MAE ↓ | MSE ↓ | SSIM ↑ | MAE ↓ | MSE ↓ | SSIM ↑ |
| ✓ | | | 32.61 | 10014.47 | 0.9735 | 0.0167 | 0.0023 | 0.9978 |
| ✓ | ✓ | | 21.71 | 2999.55 | 0.9775 | 0.0155 | 0.0025 | 0.9977 |
| ✓ | ✓ | ✓ | **16.27** | **1705.35** | **0.9866** | **0.0140** | **0.0021** | **0.9984** |

Table 2: Quantitative results of our UPFWI with different loss function settings.

Figure 5a shows the predicted velocity maps for using three loss combinations (pixel-$\ell_2$, pixel-$\ell_1\ell_2$, pixel-$\ell_1\ell_2$+perceptual) in UPFWI. The ground truth seismic data and velocity map are shown in the left column. For each loss option, we show the difference between the reconstructed and the input seismic data (on the top) and predicted velocity (on the bottom). When using pixel-wise loss in $l_2$ distance alone, there are some obvious artifacts in both seismic data (around 600 millisecond) and velocity map. These artifacts are mitigated by introducing additional pixel-wise loss in $l_1$ distance.

| Method | Marmousi | | | Salt | | |
|---|---|---|---|---|---|---|
| | MAE↓ | MSE↓ | SSIM↑ | MAE↓ | MSE↓ | SSIM↑ |
| InversionNet | 149.67 | 45936.23 | 0.7889 | 25.98 | 8669.98 | 0.9764 |
| UPFWI | 221.93 | 125825.75 | 0.7920 | 150.34 | 164595.28 | 0.7837 |

Table 3: Quantitative results of our UPFWI evaluated on Marmousi and Salt datasets.

| Network | MAE↓ | MSE↓ | SSIM↑ |
|---|---|---|---|
| CNN | 16.27 | 1705.35 | 0.9866 |
| ViT | 41.44 | 11029.01 | 0.9461 |
| MLP-Mixer | 22.32 | 4177.37 | 0.9726 |

Table 4: Quantitative results of our UP-FWI with different architectures.

With perceptual loss added, more details are correctly retained (e.g. seismic data from 400 millisecond to 600 millisecond, velocity boundary between layers). Figure 5b compares the reconstructed seismic data (in terms of residual to the ground truth) at a slice of 525 meter offset (orange dash line in Figure 5a). Clearly, the combination of pixel-wise and perceptual loss has the smallest residual.

The quantitative results are shown in Table 2. They are consistent with our observation in qualitative analysis (Figure 5a). In particular, using pixel-wise loss in $\ell_2$ distance has the worst performance. The involvement of $\ell_1$ distance mitigates velocity errors but is slightly worse on MSE and SSIM of seismic error. Adding perceptual loss boosts the performance in all metrics by a clear margin. This shows perceptual loss is helpful to retain waveform coherence, which is correlated to velocity boundary, and validates our proposed loss function (combining pixel-wise and perceptual loss).

**More Challenging Datasets:** We further evaluate our UPFWI on two more challenging tests including Marmousi and Salt (Yang & Ma, 2019) datasets and achieve solid results. For Marmousi dataset, we follow the work of Feng et al. (2021) and employ the Marmousi velocity

| $\sigma$ $(10^{-4})$ | FlatFault | | | | CurvedFault | | | |
|---|---|---|---|---|---|---|---|---|
| | PSNR | MAE↓ | MSE↓ | SSIM↑ | PSNR | MAE↓ | MSE↓ | SSIM↑ |
| 0.5 | 61.60 | 15.68 | 1343.21 | 0.9888 | 61.72 | 23.78 | 3704.00 | 0.9751 |
| 1.0 | 58.70 | 24.84 | 4010.78 | 0.9733 | 58.70 | 24.84 | 4010.78 | 0.9733 |
| 5.0 | 51.58 | 44.33 | 7592.57 | 0.9681 | 51.68 | 46.90 | 10415.38 | 0.9441 |

Table 5: Quantitative results of our UPFWI tested on seismic inputs with different noise levels.

map as the style image to construct a low-resolution dataset. Table 3 shows the quantitative results on both datasets. Although our UPFWI achieves good results on Salt dataset with preserved subsurface structures, it has clearly larger errors than the supervised InversionNet. This is due to two reasons: (a) Salt dataset has a small amount of training data (120 samples), which is very challenging for unsupervised methods; (b) the variability between training and testing samples is small, providing a significantly larger favor to supervised methods than the unsupervised counterparts. The visualization of results on Marmousi dataset and Salt data are shown in Appendix A.4.

**Other Network Architectures:** We further conducted experiments by using Vision Transformer (ViT, Dosovitskiy et al., 2020) and MLP-Mixer (Tolstikhin et al., 2021) to replace CNN as the encoder. Table 4 further shows the quantitative results. Solid results are obtained for both network architectures, indicating our proposed method is model-agnostic. Visualization results are shown in Appendix A.4.

| Missing Traces | FlatFault | | | CurvedFault | | |
|---|---|---|---|---|---|---|
| | MAE↓ | MSE↓ | SSIM↑ | MAE↓ | MSE↓ | SSIM↑ |
| 4 (5%) | 21.23 | 1772.05 | 0.9868 | 41.33 | 6914.12 | 0.9622 |
| 7 (10%) | 33.66 | 3504.25 | 0.9814 | 61.72 | 12445.90 | 0.9453 |
| 17 (25%) | 85.21 | 16731.69 | 0.9457 | 121.06 | 36770.77 | 0.8853 |

Table 6: Quantitative results of our UPFWI tested on seismic inputs with missing traces.

**Robustness Evaluation:** We validate the robustness of our UPFWI models by two additional tests: (1) testing data contaminated by Gaussian noise and (2) testing data with missing traces. The quantitative results are shown in Table 5 and Table 6, respectively. We observe that in both experiments our model is robust to a certain level of noise and irregular acquisition. Visualization results are shown in Appendix A.4.

# 6 DISCUSSION

Our UPFWI has two major limitations. Firstly, it needs further improvement on a small number of challenging velocity maps where adjacent layers have very close velocity values. We find that the lack of supervision is *not* the cause as our UPFWI yields comparable or even better results compared to its supervised counterparts. Another limitation is the speed and memory consumption for forward modeling, as the gradient of finite difference (see Equation 6) need to be stored for backpropagation. We will explore different loss functions (e.g. adversarial loss) and methods that can balance the requirement of computation resources and the accuracy in the future work. We believe the idea

of connecting CNN and PDE to solve full-waveform inversion has potential to be applied to other inverse problems with a governing PDE such as medical imaging and flow estimation.

# 7 RELATED WORK

**Physics-driven Methods:** In the past few decades, many regularization techniques have been proposed to alleviate the ill-posedness and non-linearity of FWI (Hu et al., 2009; Burstedde & Ghattas, 2009; Ramírez & Lewis, 2010; Lin & Huang, 2017; 2015b;a; Guitton, 2012; Treister & Haber, 2016). Other researchers focused on multi-scale techniques and decomposed the data into different frequency bands (Bunks et al., 1995; Boonyasiriwat et al., 2009).

**Data-driven Methods:** Recently, some researchers employed neural networks to solve FWI. Those methods can be further divided into supervised and unsupervised methods.

*Supervised:* One type of supervised methods require labeled samples to directly learn the inverse mapping, and they can be formulated as:

$$\hat{\boldsymbol{v}}(\boldsymbol{p}) = g_{\theta^*}(\boldsymbol{p}) \text{ s.t. } \theta^*(\boldsymbol{\Phi}_s) = \arg\min_{\theta} \sum_{\{\boldsymbol{v}_i, \boldsymbol{p}_i\} \in \boldsymbol{\Phi}_s} \mathcal{L}(g_\theta(\boldsymbol{p_i}), \boldsymbol{v}_i), \tag{11}$$

where $\boldsymbol{p}$ denotes the seismic measurements, $\boldsymbol{v}$ is the velocity map, $\theta$ represents the trainable weights in the inversion network $g_\theta(\cdot)$, $f(\cdot)$ is the forward modeling, and $\mathcal{L}(\cdot, \cdot)$ is a loss function. One example of supervised methods is the fully connected network proposed by Araya-Polo et al. (2018). Wu & Lin (2019) developed an encoder-decoder structured network to handle more complex velocity maps. Zhang & Lin (2020) adopted GAN and transfer learning to improve generalizability. Li et al. (2020) designed SeisInvNet to solve misaligned issue when dealing sources from different locations. In Yang & Ma (2019), a U-Net architecture was proposed with skip connections. Feng et al. (2021) proposed a multi-scale framework by considering different frequency components. Rojas-Gómez et al. (2020) developed an adaptive data augmentation method to improvegeneralizability. Sun et al. (2021) combined the data-driven and physics-based methods and proposed H-PGNN model.

Another type of supervised methods GANs to learn a distribution from velocity maps in training set as a prior (Richardson, 2018; Mosser et al., 2020). They can be formulated as:

$$\hat{\boldsymbol{v}}(\boldsymbol{z}^*) = g_{\theta^*}(\boldsymbol{z}^*) \text{ s.t. } \boldsymbol{z}^*(\boldsymbol{p}) = \arg\min_{\boldsymbol{z}} \mathcal{L}(f(g_{\theta^*}(\boldsymbol{z})), \boldsymbol{p}),$$
$$\theta^*(\boldsymbol{\Phi}_{\boldsymbol{v}}) = \arg\min_{\theta} \sum_{v_i \in \Phi_v} \mathcal{L}_{\text{GAN}}(g_\theta(\boldsymbol{\alpha_i}), \boldsymbol{v}_i), \tag{12}$$

where $\boldsymbol{\Phi}_{\boldsymbol{v}}$ is a training dataset including numerous velocity maps. $\boldsymbol{z}$ and $\boldsymbol{\alpha_i}$ are tensors sampled from the normal distribution. The iterative optimization is then performed on $\boldsymbol{z}$ to draw a velocity map sampled from the prior distribution.

*Unsupervised:* The existing unsupervised methods follow the iterative optimization paradigm and perform FWI per sample. They employ neural networks to reparameterize velocity maps. The networks serve as an implicit regularization and are required to be pretrained on an expert initial guess. Those methods can be formulated as:

$$\hat{\boldsymbol{v}}(\boldsymbol{p}) = g_{\theta^*(\boldsymbol{p})}(\boldsymbol{a}) \text{ s.t. } \theta^*(\boldsymbol{p}) = \arg\min_{\theta} \mathcal{L}(f(g_\theta(\boldsymbol{a})), \boldsymbol{p}), \tag{13}$$

where $\boldsymbol{a}$ is a random tensor. Different network architectures have been proposed including CNN-domain FWI (Wu & McMechan, 2019) and DNN-FWI (He & Wang, 2021). Zhu et al. (2021) developed NNFWI which does not need pretraining ahead, but the initial guess is still required to be fed into the PDE with estimated velocity maps.

# 8 CONCLUSION

In this study, we introduce an unsupervised method named UPFWI to solve FWI by connecting CNN and forward modeling in a loop. Our method can learn the inverse mapping from seismic data alone in an end-to-end manner. We demonstrate through a series of experiments that our UPFWI trained with sufficient amount of unlabeled data outperforms the supervised counterpart on our dataset. The ablation study further substantiates that perceptual loss is a critical component in our loss function and has a great contribution to the performance of our UPFWI.

ACKNOWLEDGMENTS

This work was supported by the Center for Space and Earth Science at Los Alamos National Laboratory (LANL), and by the Laboratory Directed Research and Development program under the project number 20210542MFR at LANL.

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

# A  APPENDIX

## A.1  NETWORK ARCHITECTURE

Since the number of receivers $R$ and the number of timesteps $T$ in seismic measurements are unbalanced ($T \gg R$), we first stack a 7×1 and six 3×1 convolutional layers (with stride 2 every the other layer to reduce dimension) to extract temporal features until the temporal dimension is close to $R$. Then, six 3×3 convolutional layers are followed to extract spatial-temporal features. The resolution is down-sampled every the other layer by using stride 2. Next, the feature map is flattened and a fully connected layer is applied to generate the latent feature with dimension 512. The decoder first repeats the latent vector by 25 times to generate a 5×5×512 tensor. Then it is followed by five 3×3 convolutional layers with nearest neighbor upsampling in between, resulting in a feature map with size 80×80×32. Finally, we center-crop the feature map (70×70) and apply a 3×3 convolution layer to output a single channel velocity map.

All the aforementioned convolutional and upsampling layers are followed by a batch normalization (Ioffe & Szegedy, 2015) and a leaky ReLU (Nair & Hinton, 2010) as activation function.

## A.2  DERIVATION OF FORWARD MODELING IN PRACTICE

Similar to the finite difference in time domain, in 2D situation, by applying the fourth-order central finite difference in space, the Laplacian of $p(\boldsymbol{r}, t)$ can be discretized as

$$
\begin{aligned}
\nabla^2 p(\boldsymbol{r}, t) =& \frac{\partial^2 p}{\partial x^2} + \frac{\partial^2 p}{\partial z^2}, \\
\approx& \frac{1}{(\Delta x)^2} \sum_{i=-2}^{2} c_i p_{x+i,z}^t + \frac{1}{(\Delta z)^2} \sum_{i=-2}^{2} c_i p_{x,z+i}^t \\
&+ O[(\Delta x)^4 + (\Delta z)^4] ,
\end{aligned}
\tag{14}
$$

where $c_0 = -\frac{5}{2}$, $c_1 = \frac{4}{3}$, $c_2 = -\frac{1}{12}$, $c_i = c_{-i}$, and $x$ and $z$ stand for the horizontal offset and the depth of a 2D velocity map, respectively. For convenience, we assume that the vertical grid spacing $\Delta z$ is identical to the horizontal grid spacing $\Delta x$.

Given the approximation in Equations 5 and 14, we can rewrite the Equation 1 as

$$
p_{x,z}^{t+1} = (2 - 5\alpha)p_{x,z}^t - p_{x,z}^{t-1} - (\Delta x)^2 \alpha s_{x,z}^t + \alpha \sum_{\substack{i=-2 \\ i \neq 0}}^{2} c_i (p_{x+i,z}^t + p_{x,z+i}^t) ,
\tag{15}
$$

where $\alpha = (\frac{v \Delta t}{\Delta x})^2$.

During the simulation of the forward modeling, the boundaries of the velocity maps should be carefully handled because they may cause reflection artifacts that interfere with the desired waves. One of the standard methods to reduce the boundary effects is to add absorbing layers around the original velocity map. Waves are trapped and attenuated by a damping parameter when propagating through those absorbing layers. Here, we follow Collino & Tsogka (2001) and implement the damping parameter as

$$
\kappa = d(u) = \frac{3uv}{2L^2} ln(R) ,
\tag{16}
$$

where $L$ denotes the overall thickness of absorbing layers, $u$ indicates the distance between the current position and the closest boundary of the original velocity map, and $R$ is the theoretical

reflection coefficient chosen to be $10^{-7}$. With absorbing layers added, Equation 6 can be ultimately written as

$$p_{x,z}^{t+1} = (2 - 5\alpha - \kappa)p_{x,z}^t - (1 - \kappa)p_{x,z}^{t-1} - (\Delta x)^2 \alpha s_{x,z}^t + \alpha \sum_{\substack{i=-2 \\ i\neq 0}}^{2} c_i(p_{x+i,z}^t + p_{x,z+i}^t) . \quad (17)$$

## A.3 OpenFWI Examples and Inversion Results of Different Methods

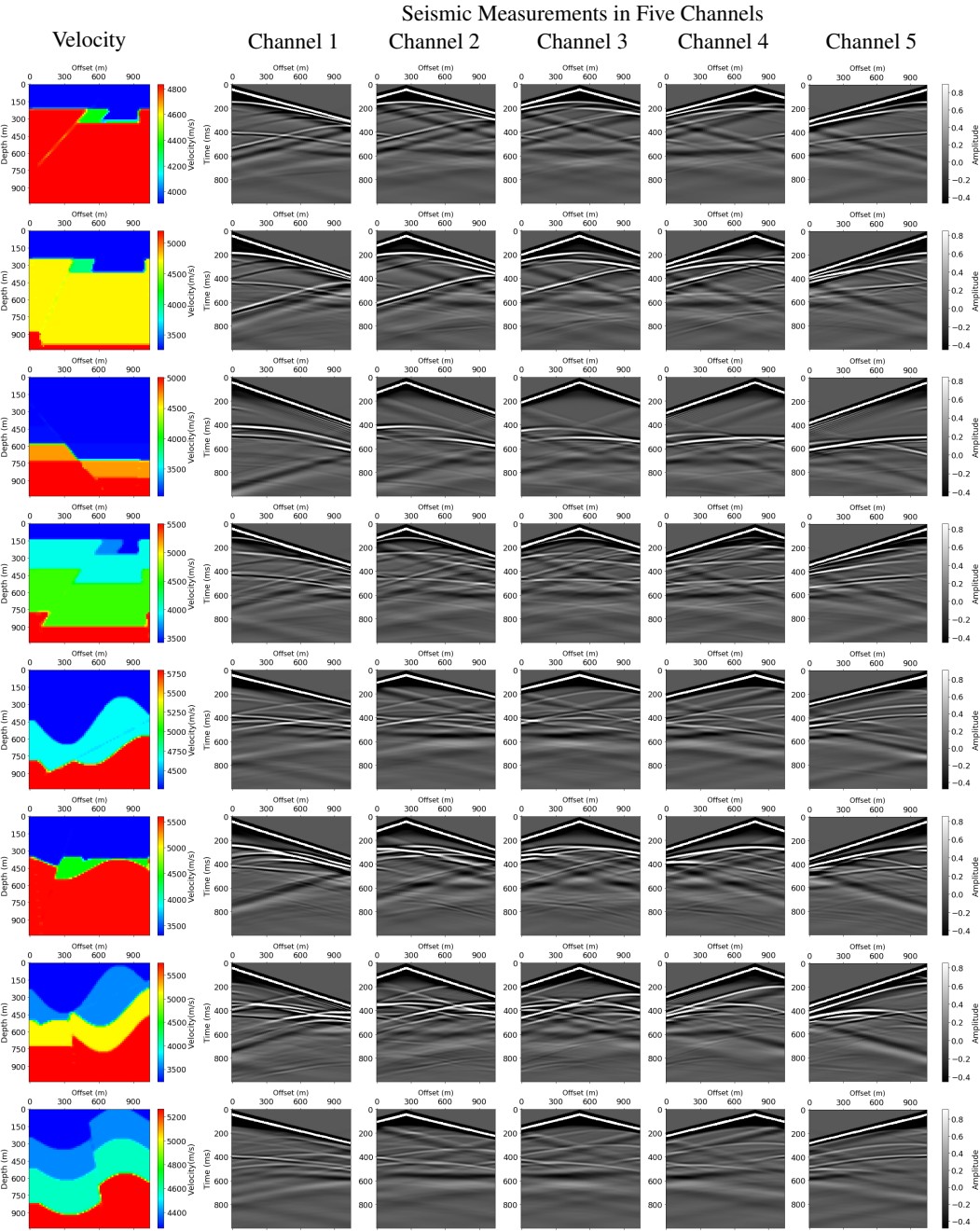

Figure 6: More examples of velocity maps and their corresponding seismic measurements in Open-FWI dataset.

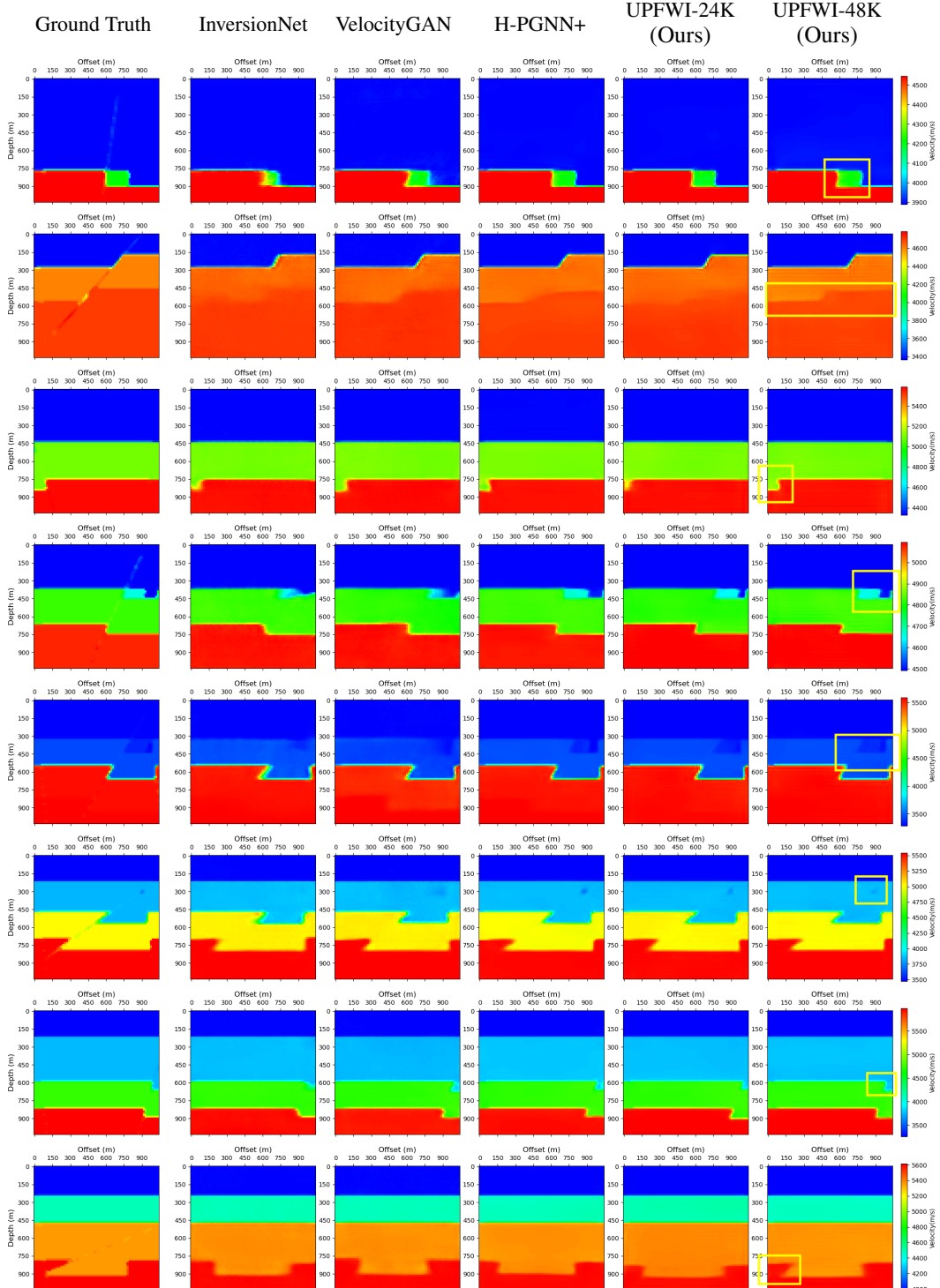

Figure 7: Comparison of different methods on inverted velocity maps of FlatFault. The details revealed by our UPFWI are highlighted.

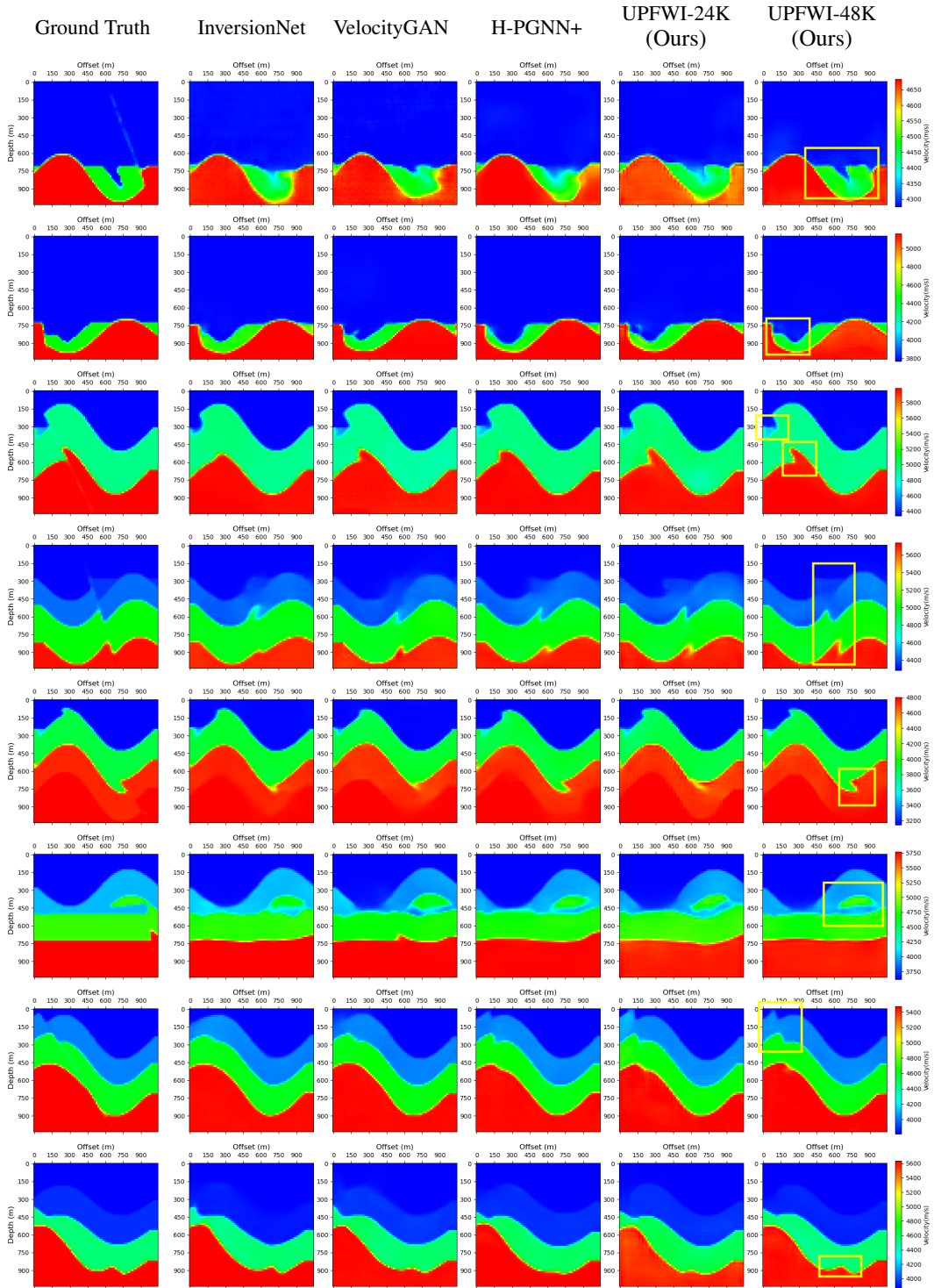

Figure 8: Comparison of different methods on inverted velocity maps of CurvedFault. The details revealed by our UPFWI are highlighted.

## A.4 ADDITIONAL EXPERIMENT RESULTS

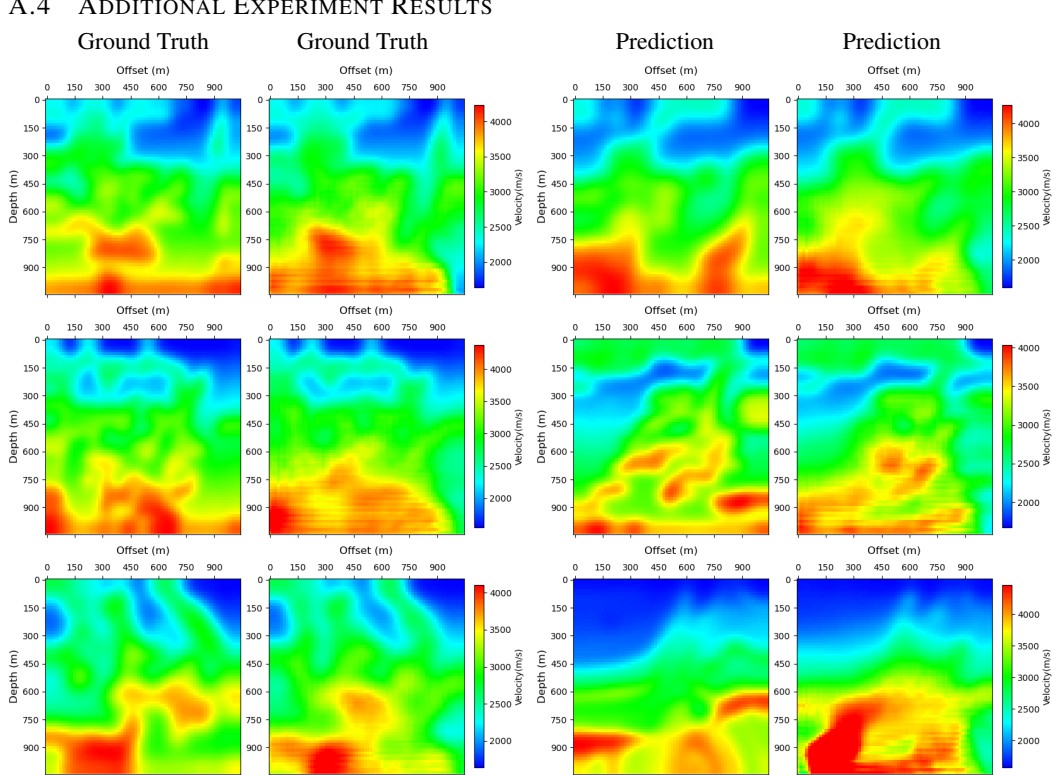

Figure 9: **Results of low-resolution Marmousi Dataset.** This dataset contains low-resolution velocity maps generated using style tranfer with the Marmousi velocity map as the style images. Our UPFWI model yields good results in shallow regions, and it also captures some geological structures in deeper regions. Similar phenomenon is also observed in the prediction of the smoothed Marmousi velocity map (bottom-right corner).

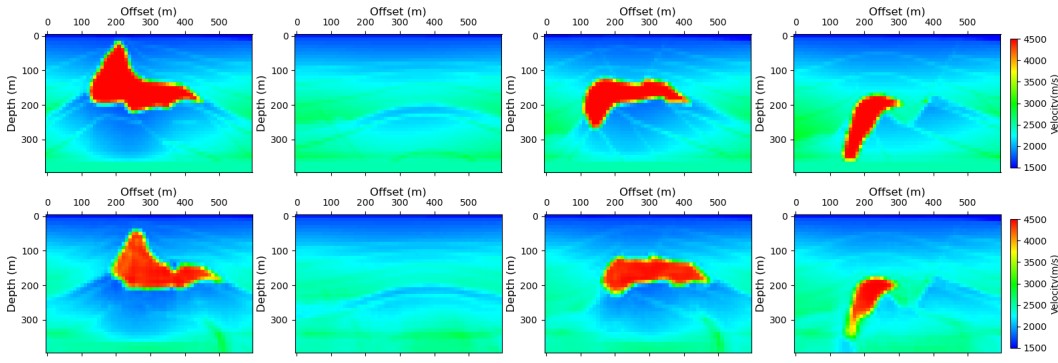

Figure 10: **Results of salt bodies dataset.** This dataset contains more complicated velocity maps. Our UPFWI model yields good velocity map prediction (bottom) on both salt bodies and background geological structures compared to the ground truth (top).

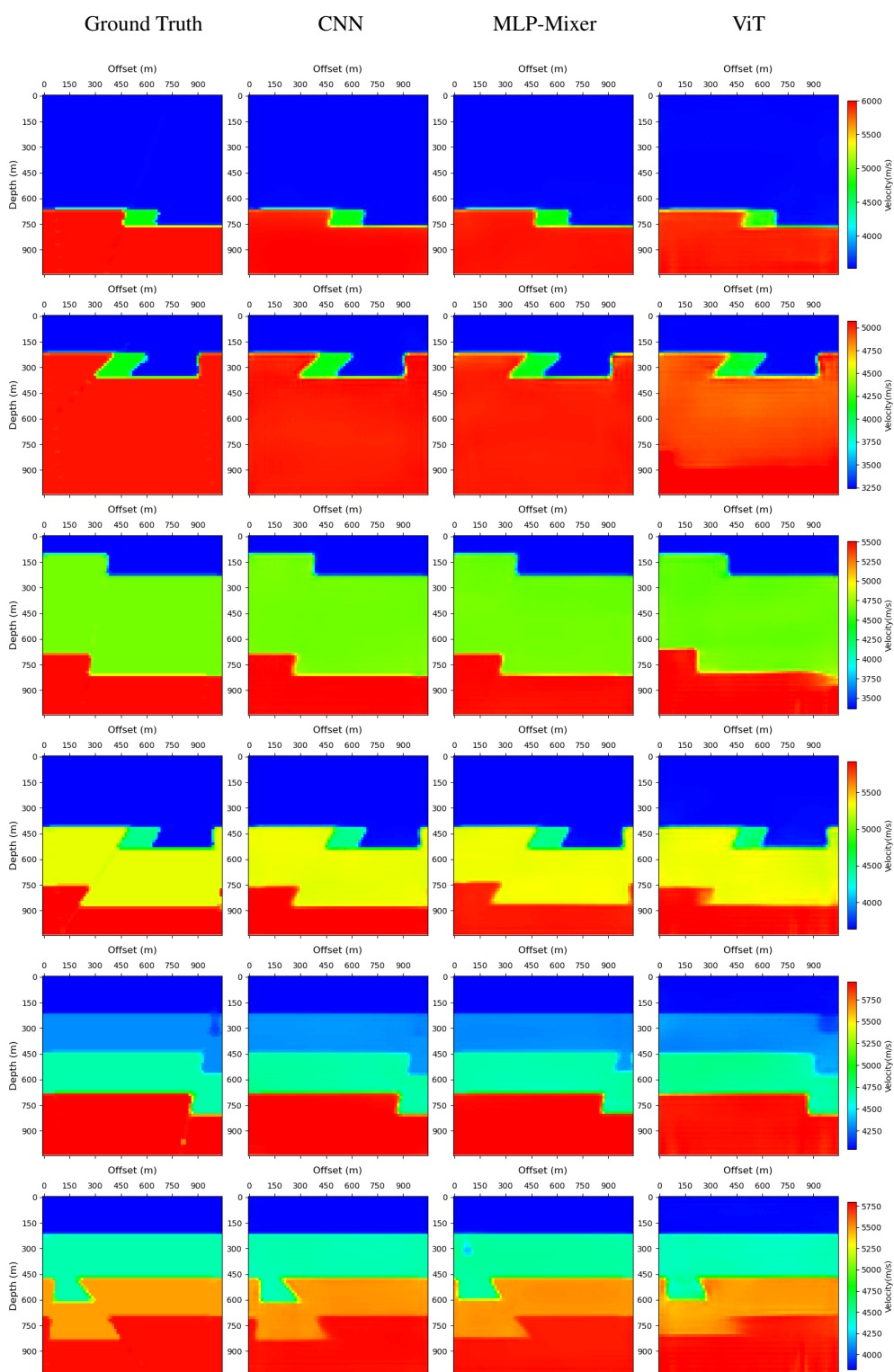

Figure 11: **Results of UPFWI with different network architectures.** We replace the CNN in our model with Vision Transformer (ViT) and MLP-Mixer as the encoder and test them on the FlatFault dataset. Both models yield reasonable velocity maps. This demonstrates that our proposed learning paradigm is model-agnostic.

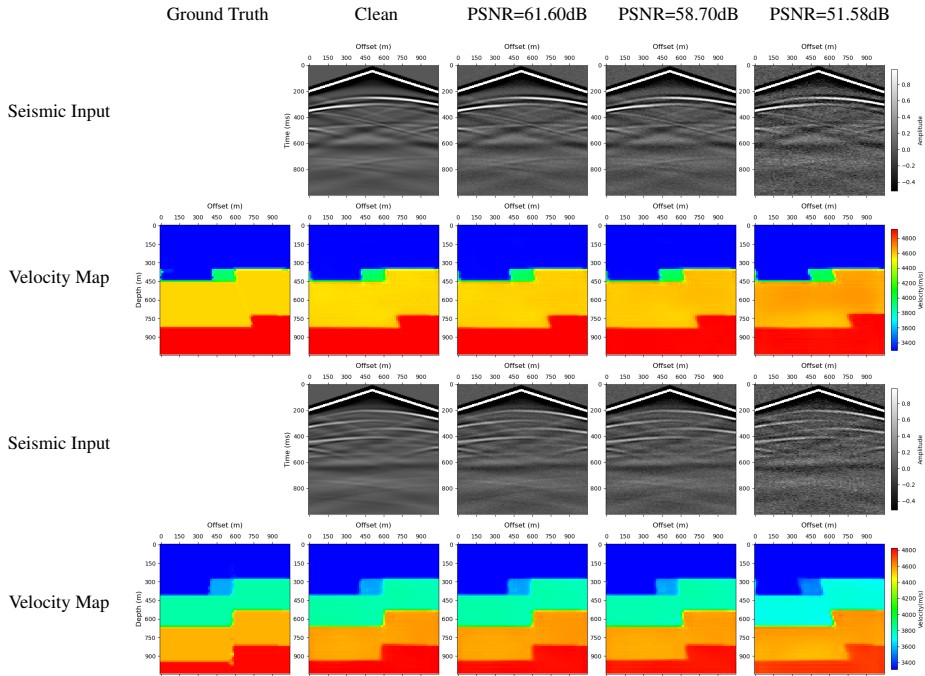

Figure 12: **Results of adding Gaussian noise to FlatFault.** The model is trained on the clean data (without noise) and tested on different levels (PSNR) of Gaussian noises. Clearly, our method is robust to the noise although slight degradation is observed when noise level increases.

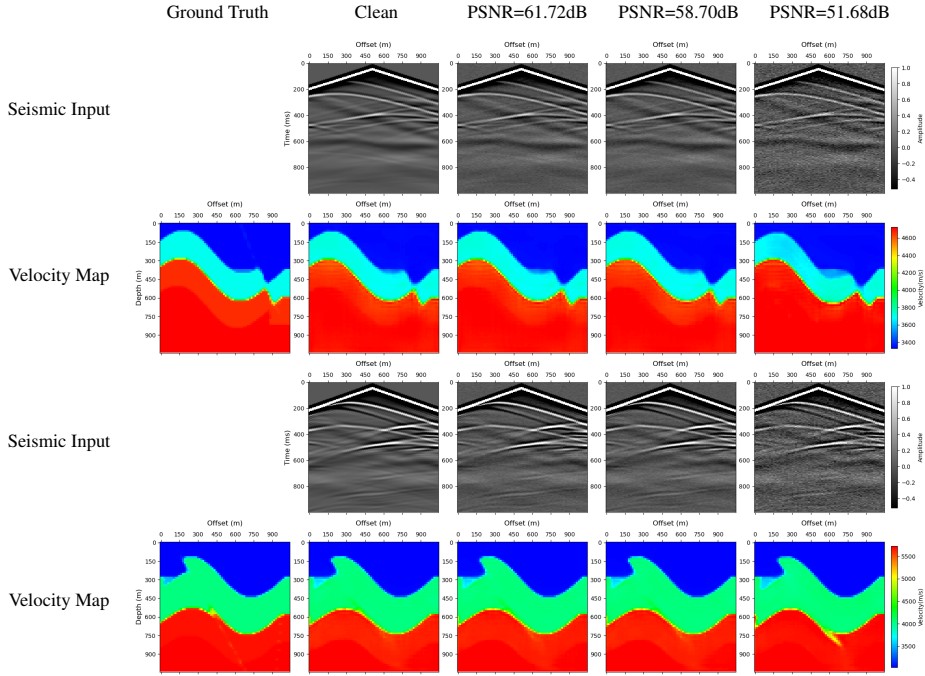

Figure 13: **Results of adding Gaussian noise to CurvedFault.** The model is trained on the clean data (without noise) and tested on different levels (PSNR) of Gaussian noises. Similar to the results of FlatFault, our method is robust to the noise although slight degradation is observed when noise level increases.

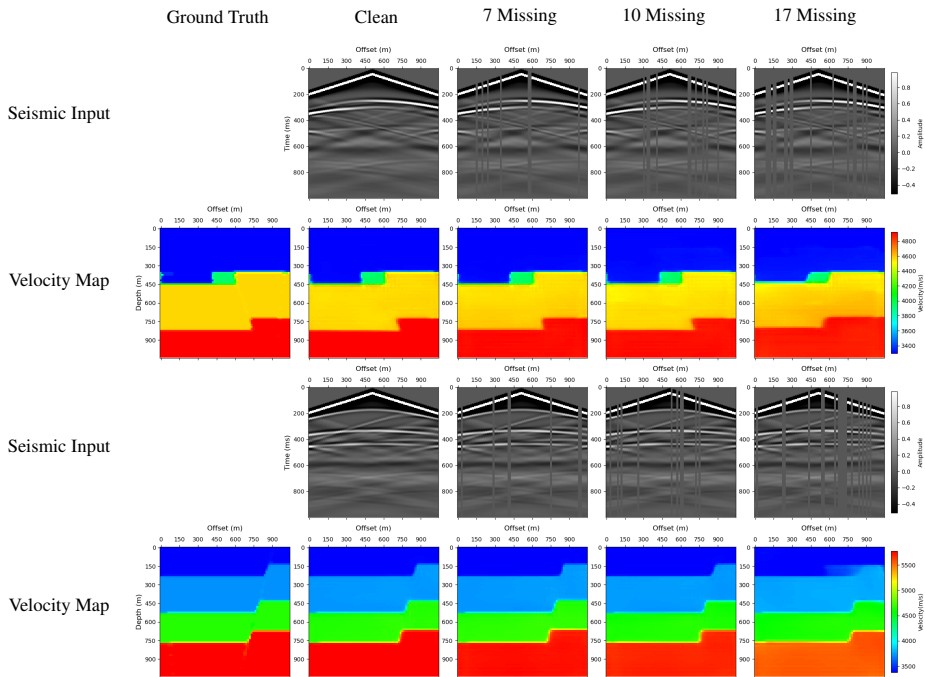

Figure 14: **Results of randomly missing traces on FlatFault.** The model is trained on the clean data (without missing traces) and tested on multiple missing rates from 5% to 25%. Our method is robust to the missing traces. Although the higher missing rate leads to shifts in velocity values, the geological structures are well preserved.

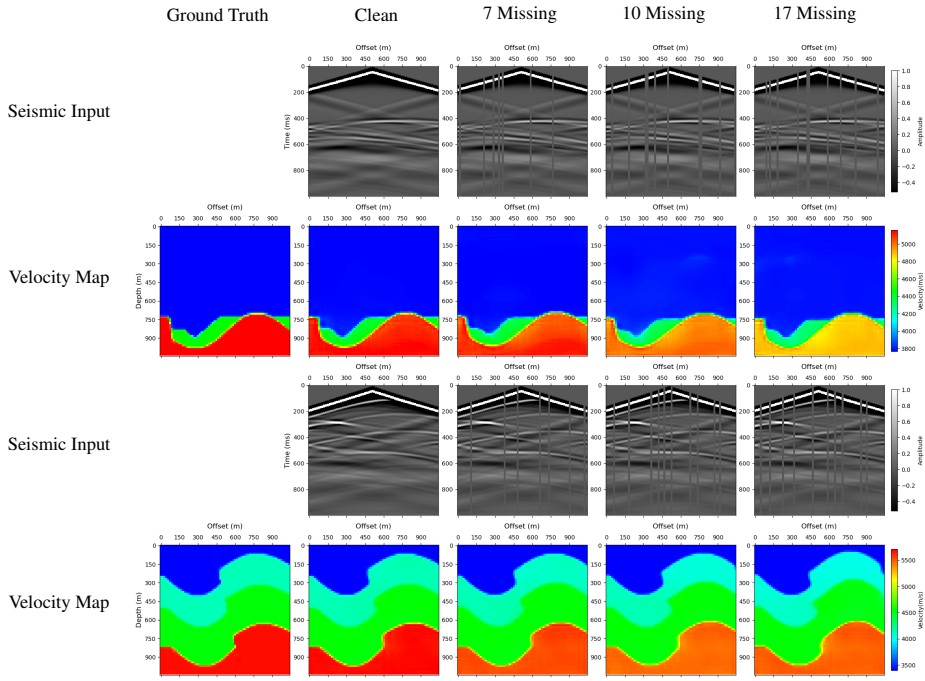

Figure 15: **Results of randomly missing traces on CurvedFault.** The model is trained on the clean data (without missing traces) and tested on multiple missing rates from 5% to 25%. Similar to the results of FlatFault, our method is robust to the missing traces. Although the higher missing rate leads to shifts in velocity values, the geological structures are well preserved.

**Additional experiments to investigate generalization.** We conducted two additional experiments: (1) training our model on the CurvedFault dataset and further testing on the FlatFault dataset (visualization results are listed in Figure 16, and quantitative results are shown in Table 7); (2) testing our model on time-lapse imaging problems (visualization results are listed in Figure 17). The results demonstrate that our proposed model yields generalization ability to a certain degree.

| Training Dataset | Test Dataset | MAE↓ | MSE↓ | SSIM↑ |
|---|---|---|---|---|
| FlatFault | FlatFault | 14.60 | 1146.09 | 0.9895 |
| CurvedFault | FlatFault | 50.80 | 17627.65 | 0.9253 |

Table 7: **Quantitative results of our UPFWI models evaluated on FlatFault.**

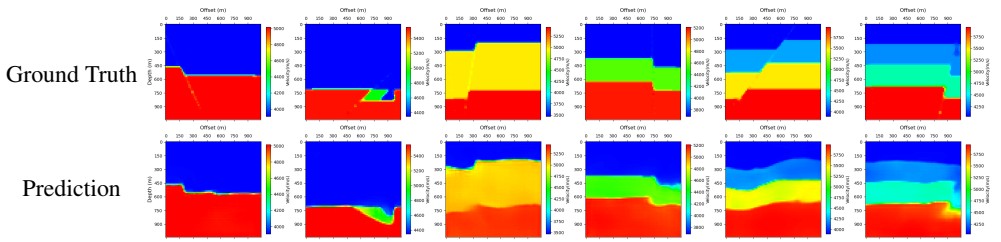

Figure 16: **Results on generalization across datasets.** The test is performed on FlatFault by applying a UPFWI model that is trained on CurvedFault dataset. Although the artifact is not negligible, the fault structures and velocity values are well preserved. This demonstrates that our model has generalizability to a certain degree.

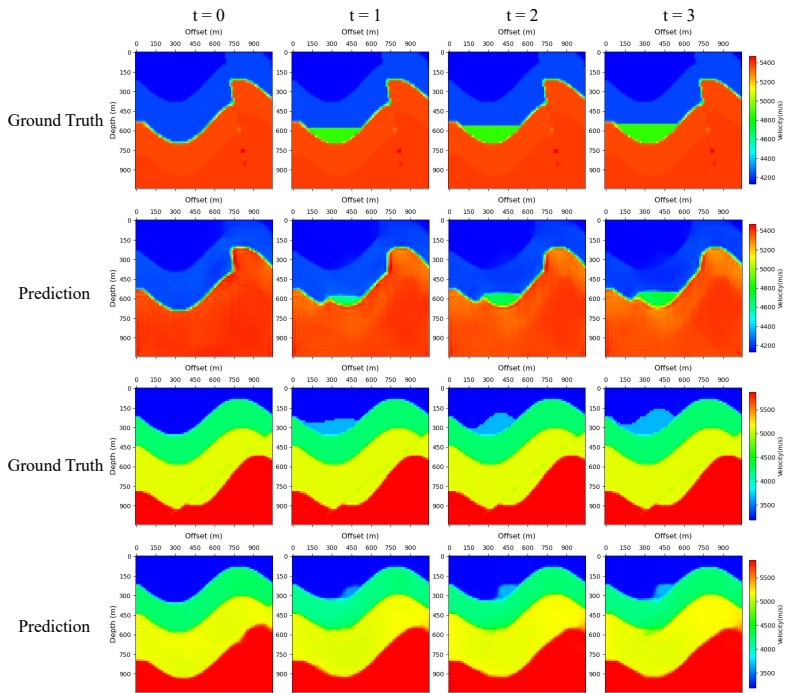

Figure 17: **Results on generalizability over geological anomalies.** The test is performed on a dataset where we add additional geological anomalies to simulate time-lapse imaging problems. The velocity maps containing those anomalies are not included during training. However, our model captures the spatial and temporal dynamics of anomalies in prediction. This demonstrates that our model has generalizability to a certain degree.

