# OpenReview forum: "Unsupervised Learning of Full-Waveform Inversion: Connecting CNN and Partial Differential Equation in a Loop"
_ICLR.cc/2022/Conference — ICLR 2022 Poster_

### Official Review · Reviewer_q6o6 · 2021-10-22

**Correctness:** 3
**Technical Novelty And Significance:** 4
**Empirical Novelty And Significance:** 4
**Recommendation:** 8
**Confidence:** 4

**Main Review:**

Overall, I find their approach very interesting. The paper is also well structured and well written. However, I have some comments and points that I feel authors could consider.

Generic comments:

- The dataset is generated for fixed measurement setup. Would this be practical or can it be extended for practical measurement procedures? They mention that physics driven approaches can be slow, but a data driven approach is very slow if a new neural network (and possibly dataset) should be generated for each measurement.

- Have authors considered extending their work for semi-supervised setting? Would labels of some samples improve results if included to the loss function?

- The improvement in their results seems to come from the fact that more samples are used (48K vs 24K). This is reasoned as this is unsupervised approach. But still it makes me wonder if somehow the other half has "better" more representative samples, i.e. more similar sample to the ground truths in their tests? To verify this, ave authors e.g. tried to switch the unlabelled part to be the labelled part (should be possible as this is simulation)?


More specific comments:

- The wave equation employed in their study assumes elastic medium and does not hold, for example, for porous materials. This is fine for this study, but could be commented.

- Introduction: it could be mentioned that OpenFWI is a simulated dataset

- Introduction, ending: MSE and SSIM values are mentioned, but especially MSE values do not have so much meaning to a reader if there is no comparison, information about typical scale etc. Relative improvement should be fine there.

- Eq (8) and (9): the loss function compares the measurement (/label) to modelled pressure signals. For "perfect models", the difference is noise. I am wondering if noise distribution could be somehow taken into account?

- It is not mentioned how dense discretization was used (spatially and temporally)? Did authors check that numerical scheme has converged (e.g. denser discretization does not change results)? Is the same discretization used both in training and test sets? Also to avoid, inverse crime type of thing, it could be beneficial to use different discretization level in training and test sets. This would avoid, for example, a case in which the neural network learns artefacts of the numerical scheme.

- Experiments: Is there any noise included to the numerical results?

- Experiments: how lambdas in Eqs (8) and (9) are chosen?

- Experiments: "batch size 16" this sounds quite small especially if 8 GPUs are used for training. Or is this per GPU batch size (in which case total batch size would be 8*16)?

- Discussion mentions "challenging velocity", which has not commented in results section. This is bit odd.


**Summary Of The Paper:**

The paper introduces a novel way to solve Full-Waveform Inversion problem which is a common problem in geological surveys. Their method is based on CNN, giving a reconstruction of velocity field for measured seismic data, and the loss function which connects CNN to a discretized version of the governing partial differential equations (the wave equation). The power of the method lies in unsupervised learning which allows one to use more data without expensive data labelling. This is demonstrated using numerical results using their simulated seismic dataset.


**Summary Of The Review:**

I am very positive about the paper and would recommend acceptance, but authors could also considers comments/points mentioned above.

---

> ### Author Response · Authors · 2021-11-23
> **Response to Reviewer q6o6 [Part 3]**
>
> **Q9: Is there any noise included to the numerical results?**
>
>
> - We conducted two additional experiments to investigate the effect of noise on our model. In the first experiment, we impose Gaussian noise with different levels to the input seismic data. The table below shows the performance of our model tested on those noisy inputs. The visualization of noisy inputs and the predicted velocity maps are provided in this link: https://tinyurl.com/vn8fhenc. We observe that our model is robust to low-level noise. As the noise level increases, more degradation occurs in performance.
>
>      Table 1: Quantitative results of our UPFWI tested on seismic inputs with different noise levels.
>
>      | Dataset | $\sigma$ ($10^{-4}$) | PSNR | MAE$\downarrow$ | MSE$\downarrow$ | SSIM$\uparrow$ |
> | :----: | :----: | :----: | :----: | :----: | :----: |
> | FlatFault | 0 | 100 | 14.60 | 1146.09 | 0.9895|
> | FlatFault | 0.5 | 61.60 | 15.68 | 1343.21 | 0.9888 |
> | FlatFault | 1.0 | 58.70 | 24.84 | 4010.78 | 0.9733 |
> | FlatFault | 5.0 | 51.58 | 44.33 | 7592.57 | 0.9681 |
> | CurvedFault |0 | 100 | 23.56 | 3639.96 | 0.9756|
> | CurvedFault |0.5 |61.72 | 23.78 | 3704.00 | 0.9751|
> | CurvedFault |1.0 | 58.70 | 24.84 | 4010.78 | 0.9733|
> | CurvedFault |5.0 | 51.68 | 46.90 | 10415.38 | 0.9441|
>
>      **(Visualization results of adding Gaussian noise: https://tinyurl.com/vn8fhenc)**
>
>
> - The second experiment focuses on the impact of missing data. To simulate irregular acquisition procedures, we randomly mask out some traces in the input seismic data. As shown in the results (see the table below and the link: https://tinyurl.com/344xnbas), our model still achieves comparable performance with irregular input, even though only full seismic data are used during training.
>
>      Table 2: Quantitative results of our UPFWI tested on seismic inputs with different numbers of missing traces.
>
>     | Dataset | Missing Traces | MAE$\downarrow$ | MSE$\downarrow$ | SSIM$\uparrow$ |
> | :----: | :----: | :----: | :----: | :----: |
> |FlatFault|0 | 14.60 | 1146.09 | 0.9895|
> |FlatFault|4 (5\%) | 21.23 | 1772.05 | 0.9868|
> |FlatFault|7 (10\%) | 33.66 | 3504.25 | 0.9814|
> |FlatFault|17 (25\%) | 85.21 | 16731.69 | 0.9457|
> |CurvedFault|0 | 23.56 | 3639.96 | 0.9756|
> |CurvedFault|4 (5\%) | 41.33 | 6914.12 | 0.9622|
> |CurvedFault|7 (10\%) | 61.72 | 12445.90 | 0.9453|
> |CurvedFault|17 (25\%) | 121.06 | 36770.77 | 0.8853|
>
>      **(Visualization results of missing traces: https://tinyurl.com/344xnbas)**
>
>
> - In summary, our model is robust to low-level Gaussian noise and irregular seismic input. The model robustness can be further improved if noisy data can be presented to the model for training.
>
>
> **Q10: How lambdas in Eqs (8) and (9) are chosen?**
>
>
> - In our experiments, we find that our model is less sensitive to $\lambda$s. For instance, the table below illustrates the choice of $\lambda_1$ (i.e., the one for $\ell_1$ pixel-wise loss). For the sake of simplicity, we choose 1 for all $\lambda$s.
>
>      Table 3: Quantitative results of our UPFWI with different hyper-parameter settings.
>
>      | $\lambda_1$ | MAE$\downarrow$ | MSE$\downarrow$ | SSIM$\uparrow$|
> | :----: | :----: | :----: | :----: |
> |0.5 | 17.43 | 2002.17 | 0.9846|
> |1.0 | 16.27 | 1705.35 | 0.9866|
> |2.0 | 17.90 | 2037.38 | 0.9847|
>
>
> **Q11: Is "batch size 16" per GPU batch size?**
>
>
> - Yes. The total batch size is 8$\times$16 (128).
>
>
> **Q12: Discussion mentions "challenging velocity", which has not commented in results section.**
>
>
> - We will comment on "challenging velocity'' in the result section. Examples of those challenging velocity maps have been provided in Figure 8 (rows 2 \& 5) listed in Appendix A.2.

---

> > ### Comment · Reviewer_q6o6 · 2021-11-23
> > **Reply**
> >
> > Thank you for your reply, what is your plan in terms to including these additional information to the actual paper?

---

> > > ### Author Response · Authors · 2021-11-23
> > > **Response to Reviewer q6o6**
> > >
> > > Thank you for your comments! We will add all those additional information in the final version of our manuscript (either in the main body or appendix, accordingly).

---

> > > ### Author Response · Authors · 2021-11-29
> > > **Response to Reviewer q6o6 [Follow Up]**
> > >
> > > We further provide our revision plan as follows:
> > > - related work section will be rewritten as (see link: https://tinyurl.com/2p82shej).
> > > - evaluations on Marmousi/Salt, evaluations on more network architectures, and evaluations on noise handling will be added to the experiment section.
> > > - the rest of the experiments and all visualization will be added to the appendix.
> > >
> > > We are happy to provide the link to the new draft if required.

---

> ### Author Response · Authors · 2021-11-23
> **Response to Reviewer q6o6 [Part 2]**
>
> **Q4: The wave equation does not hold for elastic or porous medium.**
>
>
> - In this work, we are focusing on the acoustic medium. Our proposed method has the potential to be applied to elastic or porous medium as long as the forward modeling is available, and proper regularization is provided. We will comment on it in the later version and investigate those applications in the future.
>
>
> **Q5: Introduction: it could be mentioned that OpenFWI is a simulated dataset.**
>
>
> - Yes, we will mention that OpenFWI is a simulated dataset in the introduction section. Also, as suggested by the reviewer, we will introduce more complex and realistic datasets into OpenFWI to best serve the community.
>
>
> **Q6: MSE values mentioned in Introduction do not have meaning to readers if no information about scale is provided.**
>
>
> - The velocity in all our velocity maps ranges from 3,000 meter/second to 6,000 meter/second. We will add the range of velocity to the introduction section.
>
>
> **Q7: Is noise distribution considered in the loss function?**
>
>
> - The proposed loss function takes two types of noise distribution into account. The perceptual loss implicitly considers the joint distribution of noise over multiple spatial locations while the pixel-wise loss considers each pixel independently. In particular, MSE assumes noise subject to Gaussian distribution, and MAE assumes noise subject to Laplacian distribution.
>
>
> **Q8.a: How dense discretization was used?**
>
>
> - We use the $2^{\mathrm{nd}}$ order approximation on time and $4^{\mathrm{th}}$ order approximation on space. The details of the discretization have been described in Appendix A.1.
>
>
> **Q8.b: Is the same discretization used both in training and test sets?**
>
>
> - Yes. The same discretization scheme is employed for both training and testing datasets.
>
>
> **Q8.c: Did authors check that numerical scheme has converged (e.g. denser discretization does not change results)?**
>
>
> - Yes. We checked that the numerical scheme converges. As shown in the link (https://tinyurl.com/jbuvfpej), the seismic data converge to the same result for different grid sizes. When we selected the grid sizes in the paper, both accuracy and efficiency is considered.
>
>
> **Q8.d: To avoid inverse crime, it could be beneficial to use different discretization level in training and test sets.**
>
>
> - Inverse crime is one of the main issues for any inverse problems including ours. We agree that it could be of great benefit to train the model by including different discretization scenarios and improve the robustness. However, this paper aims to develop an unsupervised framework for FWI and provide a benchmark for a fair comparison. We have already listed this as one of our future work and we will provide a brief discussion in the discussion section upon revision.

---

> ### Author Response · Authors · 2021-11-23
> **Response to Reviewer q6o6 [Part 1]**
>
> **Q1: Would this be practical or can it be extended for practical measurement procedures? A data driven approach is very slow if a new neural network (and possibly dataset) should be generated for each measurement.**
>
>
> - The proposed UPFWI framework, rather than the network trained with some specific datasets, can be extendable from fixed measurement setup to practical measurement procedures. To validate this, we ran additional experiments to simulate more practical procedures from two perspectives: (a) Marmousi \& Salt datasets to simulate more realistic subsurface, and (b) imposing noise (Gaussian noise or missing traces) to simulate more realistic measurement procedures.
>
>
> - The results (see link: https://tinyurl.com/5xbpt49u, https://tinyurl.com/5bbzr27p, https://tinyurl.com/344xnbas, and https://tinyurl.com/vn8fhenc) show that our UPFWI can be used for practical setup. However, this may introduce a problem (as the reviewer mentioned): a new network should be generated for each set of measurements. Next, we will explain why this problem is challenging to solve, how to mitigate it, and why our UPFWI is superior to physics-driven approaches.
>
>
> - It is challenging to learn a single network to map from seismic data to velocity maps universally because seismic FWI is a typical ill-posed problem. Additionally, it is infeasible to learn a unique mapping function from seismic measurements to velocity maps across the whole domain $\mathcal{D}$ that covers all subsurface structures. In this work, we have demonstrated that our UPFWI can be applied to different subdomains (FlatFault, CurvedFault \& Salt).
>
>
> - It is true that there is indeed no guarantee that a single trained network can be directly applied across different subdomains. However, there is a significant difference between our UPFWI and physics-driven approaches in data efficiency. Specifically, we consider a subdomain $\mathcal{D}_{s} \subset \mathcal{D}$ includes a collection of similar subsurface structures, and define $\Phi$ and $\Phi'$ as the training and testing dataset for the subdomain, respectively. For UPFWI, the model is trained upon $\Phi$ and tested on $\Phi'$. The major computational cost is from the training process which is related to the size of the training dataset $|\Phi|$. While for physics-driven approaches, the optimization is processed per sample/measurement $p \in \Phi'$, hence the computational cost is related to the size of testing dataset $|\Phi'|$. In real applications, $|\Phi|\ll|\Phi'|$ in general, and thus it is more efficient for UPFWI compared with other physics-driven approaches.
>
>
> - To support our claim, we further conducted two experiments to illustrate that our model trained with certain dataset can be applied to other cases directly. In the first experiment, we tested the model trained with CurvedFault on FlatFault. The table below shows the quantitative results. The visualization results (see https://tinyurl.com/thy2ak9e) show our model still yields accurate velocity value in each layer. In the other experiment, we tested our model trained with CurvedFault on time-lapse imaging data, where there is small variation over time for the same location. The growing anomaly is observed in the prediction (see https://tinyurl.com/56tatezx).
>
>      Table 1: Quantitative results of our UPFWI models evaluated on FlatFault.
>
>      |Training Dataset|Test Dataset | MAE$\downarrow$ | MSE$\downarrow$ | SSIM$\uparrow$ |
> | :----: | :----: | :----: | :----: | :----: |
> | FlatFault | FlatFault | 14.60 | 1146.09 | 0.9895 |
> | CurvedFault | FlatFault | 50.80 | 17627.65 | 0.9253 |
>
>      **(Visualization results of generalization test on FlatFault: https://tinyurl.com/thy2ak9e.)**
>
>
> **Q2: Have authors considered extending their work for semi-supervised setting? Would labels of some samples improve results if included to the loss function?**
>
>
> - Thank you very much for your suggestion. In this paper, we focused on the unsupervised setting. We will explore the semi-supervised setting in the future work.
>
>
> **Q3: Does the other half have "better" and more representative samples?**
>
>
> - We have conducted experiments to verify if there exists a bias in the other part of both FlatFault and CurvedFault datasets. The results are shown in the table below. We observe that the models trained with the other part of the dataset yield similar performance to the ones mentioned in the paper. This demonstrates that the other part of both FlatFault and CurvedFault do not have more representative samples.
>
>      Table 2: Quantitative results of our UPFWI trained on different parts of FlatFault and CurvedFault datasets.
>
>      | Dataset | First 24k | Second 24k | MAE$\downarrow$ | MSE$\downarrow$ | SSIM$\uparrow$ |
> |:----:|:----:|:----:|:----:|:----:|:----:|
> |FlatFault | $\checkmark$ | |16.27 |1705.35|0.9866|
> |FlatFault | | $\checkmark$ |16.78 |1796.91|0.9864|
> |CurvedFault | $\checkmark$ | |29.59|5712.25|0.9652|
> |CurvedFault | | $\checkmark$ |28.12|5377.52|0.9666|

---

### Official Review · Reviewer_ZRwQ · 2021-11-01

**Correctness:** 2
**Technical Novelty And Significance:** 2
**Empirical Novelty And Significance:** 2
**Recommendation:** 3
**Confidence:** 5

**Main Review:**

Strength:
Propose an unsupervised formulation for FWI

Weakness:

**(1) Sim2Real gap between OpenFWI and seismic inversion:**

OpenFWI obtains the velocity map and raw shot signals through simulation.  In this simulation, OpenFWI assumes uniform acquisition geometry, where the receivers are positioned at each grid with an interval of 15 meters. In real seismic inversion for exploration, the acquisition geometry may be non-uniform. Moreover, dataset such as (Yang & Ma, 2019) includes velocity maps with more complicated salt bodies.  It seems that OpenFWI provides more simplified velocity maps.  Therefore, it remains unknown how models trained on OpenFWI generalize to real scenarios.

**(2) Convolution on raw seismic traces:**

As shown in the paper, the raw seismic traces could be simulated through forward modeling of PDE. Under the irregular acquisition geometry, the raw seismic traces are not image-like as we do not have receivers in a grid. In these scenarios, convolution operations that require matrix input could not be applied. Therefore, the convolution networks may be restricted to synthetic datasets.

**(3) The role of PDE forward modeling:**

It seems that the PDE forward modeling is regarded as a blackbox during unsupervised training. Is there any illustration of how the forward modeling helps the training of CNN?

**(4)The novelty of the unsupervised learning paradigm:**

Is the proposed learning paradigm model-agnostic? How does it perform when we switch the backbone model to transformer (Dosovitskiy et al 2020) or MLP (Tolstikhin et al 2021)?




Fangshu Yang and Jianwei Ma. Deep-learning inversion: A next-generation seismic velocity model building method. Geophysics, 84(4):R583–R599, 2019.

Dosovitskiy A, Beyer L, Kolesnikov A, et al. An image is worth 16x16 words: Transformers for image recognition at scale[J]. arXiv preprint arXiv:2010.11929, 2020.

Tolstikhin I, Houlsby N, Kolesnikov A, et al. Mlp-mixer: An all-mlp architecture for vision[J]. arXiv preprint arXiv:2105.01601, 2021.


**Summary Of The Paper:**

This article studies the Full-Waveform Inversion (FWI) problem. The major contributions of this article could be summarized as

(1) Formulate the FWI in an unsupervised way

(2) Propose a learning strategy with perceptual loss

(3) Introduce a dataset for benchmark


**Summary Of The Review:**

The topic is of general interest and the paper is mostly well written. However, there exist concerns about a lack of novelty and incremental improvements.

The authors are strongly encouraged to address these shortcomings by:

i) provide a better justification of the OpenFWI dataset, especially on how OpenFWI bridges the gap between simulation and real seismic inversion

ii) motivate the idea Neural Network+PDE  better

iii) improve the experimental section by studying if the proposed method is model-agnostic

---

> ### Author Response · Authors · 2021-11-23
> **Response to Reviewer ZRwQ [Part 3]**
>
> **Q4: Is the proposed learning paradigm model-agnostic? How does it perform when we switch the backbone model to transformer (Dosovitskiy et al 2020) or MLP (Tolstikhin et al 2021)?**
>
> Our proposed learning paradigm is model-agnostic. To verify this, we have conducted additional experiments on different network architectures. Following your suggestions, we replace CNN with Vision Transformer (ViT) and MLP-Mixer as the encoder, respectively. The models are trained on the FlatFault dataset using 24K unlabeled samples. The comparison between different architectures on inverted velocity maps is listed in this link https://tinyurl.com/9bppamzs. The table below shows the quantitative results. Due to limited time, we have not fine-tuned the hyperparameters. However, it is observed that both ViT and MLP-Mixer yield reasonable inversion results. In addition, ViT performs slightly worse than MLP-Mixer and CNN-based UPFWI. The potential reason is that transformers rely more on data augmentation techniques [7] which cannot be directly applied to seismic data and velocity maps due to their physics meanings. In summary, this demonstrates that our proposed learning paradigm is model-agnostic.
>
> Table 3. Quantitative results of our UPFWI with different network architectures.
>
> |Network Architecture | MAE$\downarrow$ | MSE$\downarrow$ | SSIM$\uparrow$ |
> |---|---|---|---|
> |CNN | 16.27 | 1705.35 | 0.9866 |
> |ViT | 41.44 | 11029.01 | 0.9461 |
> |MLP-Mixer | 22.32 | 4177.37 | 0.9726 |
>
> **(Visualization of results based on ViT and MLP-Mixer: https://tinyurl.com/9bppamzs.)**
>
> [7] Training data-efficient image transformers \& distillation through attention, ICML, 2021

---

> ### Author Response · Authors · 2021-11-23
> **Response to Reviewer ZRwQ [Part 2]**
>
> **Q2: Convolution networks may be restricted to synthetic datasets due to irregular acquisition geometry.**
>
> This is an important point. Dealing with irregular acquisition geometry is a common challenge for both physics-based approaches and data-driven methods including ours. Due to its great importance, various data pre-processing techniques have been developed to overcome this issue. Just to name a few, interpolation approaches based on mathematical transform [4][5]; prediction filter-based methods [6]. Those data pre-processing techniques would alleviate the irregular acquisition issue by converting irregular spatial sampling into a regular one, and thus allow us to further employ our proposed technique.
>
> Although this topic is out of the main focus of our current work, we still conduct a robustness test to validate how our proposed model would perform under irregular acquisition. We simulate the procedure by randomly masking out 5\% to 25\% traces in input seismic data. The tested model is trained using data without missing traces. Examples of input seismic data and predicted velocity maps are listed in this link https://tinyurl.com/344xnbas. The table below shows the quantitative results. We observe that although a higher missing rate leads to shifts in velocity values, the geological structures are still well preserved. This demonstrates the robustness of our model on irregular inputs.
>
> Table 2. Quantitative results of our UPFWI tested on seismic inputs with different numbers of missing traces.
>
>  |Dataset | Missing Traces | MAE$\downarrow$ | MSE$\downarrow$ | SSIM$\uparrow$|
> |---|---|---|---|---|
> |FlatFault | 0 | 14.60 | 1146.09 | 0.9895|
> |FlatFault|4 (5\%) | 21.23 | 1772.05 | 0.9868|
> |FlatFault| 7 (10\%) | 33.66 | 3504.25 | 0.9814 |
> |FlatFault|17 (25\%) | 85.21 | 16731.69 | 0.9457|
> | | | | | | |
> |CurvedFault| 0 | 23.56 | 3639.96 | 0.9756|
> |CurvedFault|4 (5\%) | 41.33 | 6914.12 | 0.9622|
> |CurvedFault|7 (10\%) | 61.72 | 12445.90 | 0.9453|
> | CurvedFault |17 (25\%) | 121.06 | 36770.77 | 0.8853|
>
> **(Visualization results of missing traces: https://tinyurl.com/344xnbas.)**
>
> **Q3: It seems that the PDE forward modeling is regarded as a blackbox during unsupervised training. Is there any illustration of how the forward modeling helps the training of CNN?**
>
> Thank you very much for pointing this out. The PDE forward model in our framework is not a black box. Instead, the explicit mathematical solver of the acoustic-wave equation is used in our framework. This implementation can be considered as a differentiable operator without learnable weights. With the PDE forward modeling incorporated, we can further obtain the simulated seismic data using the intermediate output (velocity map) from the CNN. This allows us to lift the label requirement on velocity, thus converting the problem from supervised to unsupervised.
>
> The PDE forward model plays a critical role of regularization to constrain the outputs from the CNN to be physically meaningful velocity maps. To validate the effectiveness of different implementation strategies (i.e., black box versus ours), we have conducted experiments by replacing the PDE forward model with a trainable CNN and find that the proposed framework turns itself into an auto-encoder. It means the velocity prediction can be an arbitrary feature map as long as the decoder can reconstruct the input seismic data.
>
> [4] Dreamlet-based interpolation using POCS method, JAG, 2014
>
> [5] 3D interpolation of irregular data with a POCS algorithm, Geophysics, 2006
>
> [6] Multistep autoregressive reconstruction of seismic records, Geophysics, 2007

---

> ### Author Response · Authors · 2021-11-23
> **Response to Reviewer ZRwQ [Part 1]**
>
> **Q1: Sim2Real gap between OpenFWI and seismic inversion:**
>
> **Q1.a: OpenFWI assumes uniform acquisition geometry. However, the acquisition geometry may be non-uniform in real scenarios.**
>
> Although we provide seismic data under the uniform acquisition setting in OpenFWI, it is easy to transform the provided data into non-uniform acquisition setting. For example, masking out some traces in the seismic data can be applied to simulate non-uniform distances between receivers.
>
> **Q1.b: OpenFWI provides more simplified velocity maps compared to other datasets such as the salt dataset (Yang \& Ma, 2019). Therefore, it remains unknown how models trained on OpenFWI generalize to real scenarios.**
>
> Based on the reviewer's recommendation, we have included more challenging datasets into our OpenFWI to mitigate the gap between simulation and real scenarios. Particularly, we create Marmousi dataset (24K pairs of seismic data and velocity maps) following [3].
>
> **Q1.c: How does the method generalize to real scenario.**
>
> We have conducted two experiments to evaluate our proposed UPFWI on more physically realistic datasets.
>
> - **Experiment 1 -- Salt:** In this experiment, we applied our method to Salt dataset [1] which contains velocity maps with more complicated salt bodies. The visualization results are listed in this link https://tinyurl.com/5xbpt49u. The table below shows the quantitative results of our UPFWI and InversionNet [2]. Although our UPFWI achieves good results on Salt dataset with preserved subsurface structures, it has clearly larger errors than the supervised InversionNet. This is due to two reasons: (a) Salt dataset has a small amount of training data (120 samples), which is very challenging for unsupervised methods; (b) the variability between training and testing samples is small, providing a significantly larger favor to supervised methods than the unsupervised counterparts.
> - **Experiment 2 -- Marmousi:** In this experiment, we followed the work of [3] and employed the Marmousi velocity map as the style image to construct a low-resolution dataset with more physically realistic subsurface velocity maps. 48K unlabeled samples are used to train our UPFWI model. The examples of predicted velocity maps in the test set are shown in this link https://tinyurl.com/5bbzr27p. The table below shows the quantitative results of our UPFWI and InversionNet [2]. Our model yields good results in shallow regions and captures some geological structures in deeper regions. A similar phenomenon is also observed in the prediction of the smoothed Marmousi velocity map.
>
>    Table 1. Quantitative results evaluated on Marmousi and Salt.
>     |Dataset | Model | MAE$\downarrow$ | MSE$\downarrow$ | SSIM$\uparrow$|
> |---|---|---|---|---|
> |Salt|InversionNet | 25.98 | 8669.98 | 0.9764 |
> |Salt| UPFWI | 150.34 | 164595.28 | 0.7837 |
> |Marmousi|InversionNet|149.67 | 45936.23 | 0.7889|
> |Marmousi|UPFWI| 221.93 | 125825.75 | 0.7920|
>
>     **(Visualization results of Salt: https://tinyurl.com/5xbpt49u.)**
>
>     **(Visualization results of Marmousi: https://tinyurl.com/5bbzr27p.)**
>
> The results in those two experiments demonstrate that our UPFWI method can be applied to physically realistic velocity maps.
>
> [1]  Deep-learning inversion: A next-generation seismic velocity model building method, Geophysics, 2019
>
> [2] InversionNet: An efficient and accurate data-driven full waveform inversion. IEEE TCI, 2019
>
> [3] Multiscale Data-driven Seismic Full-waveform Inversion with Field Data Study, IEEE TGRS, 2021

---

> ### Author Response · Authors · 2021-11-30
> **Response to Reviewer ZRwQ [Follow Up]**
>
> Dear reviewer, we politely request your feedback on our response. We would love to provide additional information and answer your questions. Thank you very much.

---

> ### Comment · Reviewer_ZRwQ · 2021-11-30
> **Response**
>
> I would like to express my sincere appreciation for the additional experiments and explanations.  The extensive evaluation illustrates the performance of the model in different aspects. However, there are several concerns raised in the updated version of this paper.
>
> **The novelty of OpenFWI dataset**
>
> Based on the previous sim2real gap mentioned in the review, the authors include salt [1] and  Marmousi [2] datasets into the OpenFWI dataset. Although the effort is appreciated, it comes with a concern about OpenFWI's novelty. What makes OpenFWI different than the combination of existing datasets?
>
> **Masking experiments**
>
> I would like to thank the authors for the masking experiments. However, there are still some concerns raised from the experiments. (1) Does the masking procedure set some traces to zeros and keep the original shape of the image-like data? If that is the case, the performance degradation is predictable as we give the network the wrong information about the raw seismic signals. It would be better to regard seismic traces as location-aware 1D arrays. That is the major reason for questioning convolution in FWI. (2) In the robust experiments, the test data still have full traces. As a fair comparison, it would be better to also mask the input when we perform the evaluation.
>
>
> **PDE as a white box**
>
> Thanks to the author for the clarification. Are there any experiment results in the updated version that clarifies the advantages of PDE versus black box CNN?  Moreover, how is the gradient flow in the UPFWI? Does PDE pass gradients to the covolutional network?
>
>
> [1] Deep-learning inversion: A next-generation seismic velocity model building method, Geophysics, 2019
>
> [2] Multiscale Data-driven Seismic Full-waveform Inversion with Field Data Study, IEEE TGRS, 2021

---

> > ### Author Response · Authors · 2021-12-03
> > **Response to Review ZRwQ**
> >
> > **Q1: The novelty of OpenFWI dataset -- The authors include Salt and Marmousi datasets into the OpenFWI dataset. What makes OpenFWI different than the combination of existing datasets?**
> >
> > Let us first clarify that the novelty of our submission is *not* the OpenFWI dataset but our unsupervised method (***UPFWI***) that combines CNN and PDE in a loop. To make our contribution clear, *the UPFWI method is our key contribution while the OpenFWI dataset is supportive of it*. The purpose of the OpenFWI dataset is to encourage the study of unsupervised learning of FWI. It is large, which is important for unsupervised learning. It covers two subsurface structures (flat, curved), allowing us to study the generalizability of models.  Thanks for the reviewer's recommendation, adding more challenging data (e.g. Marmousi) is more helpful to serve this purpose by enriching the evaluation. The experiments on the OpenFWI dataset show that our UPFWI leverages a large amount of unlabeled data to achieve accurate prediction for different subsurface structures. Furthermore, our UPFWI model trained on CurvedFault alone yields generalizability to FlatFault. We hope this provides a good baseline for the research community.
> >
> >
> > **Q2. Masking experiments:**
> >
> > **Q2.a: Does the masking procedure set some traces to zeros and keep the original shape of the image-like data?**
> >
> > Yes, we set some traces to zeros and keep the original shape.
> >
> > **Q2.b: It would be better to regard seismic traces as location-aware 1D arrays. That is the major reason for questioning convolution in FWI.**
> >
> > Good point! We follow your suggestion and conduct an additional experiment to treat seismic data as location-aware 1D arrays. Specifically, we replace CNN encoder with a transformer where each seismic trace is considered as a token, and its location is encoded by position embedding. The experiment is running and will take a couple of days. We will update the experiment results by the end of this week.
> >
> > **Q2.c: In the robust experiments, the test data still have full traces. As a fair comparison, it would be better to also mask the input when we perform the evaluation.**
> >
> > We clarify that masked traces (rather than full traces) are used for testing.
> >
> > **Q3: PDE as a white box:**
> >
> > **Q3.a: Are there any experiment results in the updated version that clarifies the advantages of PDE versus black box CNN?**
> >
> > Yes. Please refer to this link (https://tinyurl.com/5bwvncbr) for visualization results of replacing PDE with a black box CNN. The input seismic data is well reconstructed. However, the model fails to predict velocity maps due to the lack of constraints.
> >
> > **Q3.b: Moreover, how is the gradient flow in the UPFWI? Does PDE pass gradients to the convolution network?**
> >
> > Yes, the PDE passes gradients to the convolution network since the PDE is differentiable.

---

> > ### Author Response · Authors · 2021-12-06
> > **Response to Review ZRwQ [Update]**
> >
> > Due to similar reasons we have discussed in ViT, the model needs more time to converge, but we would like to share some preliminary results first. The visualization results are listed in this link: https://tinyurl.com/2p8p44me, and the table below shows quantitative results. We observe that our model with a transformer as the encoder still yields reasonable velocity maps. It demonstrates that seismic data can be treated as location-aware 1D arrays in our UPFWI.
> >
> > Table 1: Quantitative results of our UPFWI with Transformer as the encoder.
> >
> > | Network Architecture | MAE$\downarrow$ | MSE$\downarrow$ | SSIM$\uparrow$ |
> > | :-: | :-: | :-: | :-: |
> > |Transformer | 54.60 | 13204.71 | 0.9343 |
> >
> > **(Visualization results of Transformer: https://tinyurl.com/2p8p44me.)**

---

### Official Review · Reviewer_vGXF · 2021-11-02

**Correctness:** 3
**Technical Novelty And Significance:** 2
**Empirical Novelty And Significance:** 2
**Recommendation:** 5
**Confidence:** 4

**Main Review:**

**Strengths**
- Combining the CNN with the physics-based model helps to alleviate the need for ground truth velocity maps, which is convenient.

- The authors introduce a new large-scale dataset which could be of interest to the community working on data-driven solutions to full-waveform inversion (FWI).

- The authors compare to other learning-based approaches and perform an ablation study to evaluate their proposed method.

**Weaknesses**

- There is already a very large body of work on combining neural networks (e.g., CNNs) and physics-based models for solving PDEs, including for FWI. Yet, in the introduction the authors cite only a single paper to put their work in context. I recognize that there is a more extensive review in the related work (at the end), but reading the paper I was confused if the authors are actually claiming the method as a contribution or if they are only applying it to FWI.

- Indeed there are a few papers that ought to be cited which also combine a neural network with the physics-based model for FWI. For example, Mosser et al. (2020) combine a generative model with a physics-based model in order to improve FWI inversion, and He and Wang propose a very similar approach that uses a CNN to output the velocity field, and then they optimize the CNN through the physics-based forward model. This is actually extremely close to what the authors propose; the main difference seems to be that He and Wang perform test-time optimization to overfit to a single scene (so also unsupervised) while the authors propose to train across a dataset. This works should also be discussed, and the authors should clarify what the novelty of the proposed approach is.

    He, Qinglong, and Yanfei Wang. "Reparameterized full-waveform inversion using deep neural networks." Geophysics 86.1 (2021): V1-V13.

    Mosser, Lukas, Olivier Dubrule, and Martin J. Blunt. "Stochastic seismic waveform inversion using generative adversarial networks as a geological prior." Mathematical Geosciences 52.1 (2020): 53-79.

- As with any other learned approach, generalization is likely a challenge with the proposed method. How does the method perform on slightly different datasets? For example can the model that is trained on CurvedFault generalize to FlatFault? Or can these models generalize to standard FWI datasets like the Marmousi dataset or the 2004 BP model (ref below)? These are standard datasets and so these results would likely be of interest to the FWI community.

    Billette, F. J., and Sverre Brandsberg-Dahl. "The 2004 BP velocity benchmark." 67th EAGE Conference & Exhibition. European Association of Geoscientists & Engineers, 2005.

**Typos/Misc.**


Section 1
- It should be clarified that the proposed dataset is a simulated 2D dataset.

- It's not clear what supervised method is being referred to here: "...26.77% smaller than that of the supervised method"

- Full Waveform Inversion or Full-Waveform Inversion? Both are used (title, intro text, Section 2 header)

Section 3

"Physics-informed", but in section 1 "Physical-informed"


**Summary Of The Paper:**

The paper presents a method for full-waveform inversion (an inverse problem in seismic imaging) that combines a convolutional neural network (CNN) with a physics-based forward modeling operator. As opposed to other work which directly learns the inverse mapping from measurements to the sought-after velocity parameter, the authors propose an "unsupervised" approach. Here, the CNN takes the seismic measurements as input and predicts a velocity field. Then, the physics-based forward operator produces measurements which can be compared to the original measurements to train the network. The authors evaluate their method on a new simulated dataset and show that it can outperform other supervised approaches.


**Summary Of The Review:**

Overall I lean slightly negative on the paper. My main criticism is that the novelty of the approach seems to be low. Moreover, the closest related work that I am aware of does not appear to be cited or discussed. Finally, while the method performs well in simulation, there are no results that clearly evaluate generalization and/or performance on more challenging datasets. I would be open to revising my score if the authors can clearly address these concerns.

---

> ### Author Response · Authors · 2021-11-23
> **Response to Reviewer vGXF [Part 2]**
>
> **Q2: How does the method perform on slightly different datasets? Can these models be generalized to standard FWI datasets like the Marmousi dataset or the 2004 BP model?**
>
>
> - Following your suggestions, we have conducted three additional experiments as follows.
>
>
> - **Experiment 1 -- Generalizability:** In this experiment, the test is performed on FlatFault by applying our UPFWI model trained on CurvedFault dataset. The examples of predicted velocity maps are listed in this link: https://tinyurl.com/thy2ak9e. The table below shows the quantitative results. Although the artifact is not negligible, the fault structures and velocity values are well preserved. This demonstrates that our model has generalizability to a certain degree.
>
>      Table 1: Quantitative results of our UPFWI models evaluated on FlatFault.
>
>      | Training Dataset | Test Dataset | MAE$\downarrow$ | MSE$\downarrow$ | SSIM$\uparrow$ |
> | :----: | :----: | :----: | :----: | :----: |
> | FlatFault | FlatFault | 14.60 | 1146.09 | 0.9895 |
> | CurvedFault | FlatFault | 50.80 | 17627.65 | 0.9253 |
>
>      **(Visualization results of generalization test on FlatFault: https://tinyurl.com/thy2ak9e.)**
>
>
> - **Experiment 2 -- Marmousi:** In this experiment, we followed the work of [4] and employed the Marmousi velocity map as the style image to construct a low-resolution dataset with more physically realistic subsurface velocity maps. 48K unlabeled samples are used to train our UPFWI model. The examples of predicted velocity maps in the test set are shown in this link: https://tinyurl.com/5bbzr27p. The table below shows the quantitative results of our UPFWI and InversionNet [5]. Our model yields good results in shallow regions and captures some geological structures in deeper regions. A similar phenomenon is also observed in the prediction of the smoothed Marmousi velocity map.
>
>     Table 2. Quantitative results evaluated on Marmousi.
>
>      | Model | MAE$\downarrow$ | MSE$\downarrow$ | SSIM$\uparrow$ |
> | :----: | :----: | :----: | :----: |
> | InversionNet | 149.67 | 45936.23 | 0.7889 |
> | UPFWI | 221.93 | 125825.75 | 0.7920 |
>
>      **(Visualization results of Marmousi: https://tinyurl.com/5bbzr27p.)**
>
>
> - **Experiment 3 -- Salt:** In this experiment, we applied our method to Salt dataset [6] which contains velocity maps with more complicated salt bodies. The visualization results are listed in this link: https://tinyurl.com/5xbpt49u. The table below shows the quantitative results of our UPFWI and InversionNet [5] . Although our UPFWI achieves good results on Salt dataset with preserved subsurface structures, it has clearly larger errors than the supervised InversionNet. This is due to two reasons: (a) Salt dataset has a small amount of training data (120 samples), which is very challenging for unsupervised methods; (b) the variability between training and testing samples is small, providing a significantly larger favor to supervised methods than the unsupervised counterparts.
>
>      Table 3. Quantitative results evaluated on Salt.
>
>      | Model | MAE$\downarrow$ | MSE$\downarrow$ | SSIM$\uparrow$ |
> | :----: | :----: | :----: | :----: |
> | InversionNet | 25.98 | 8669.98 | 0.9764 |
> | UPFWI | 150.34 | 164595.28 | 0.7837 |
>
>      **(Visualization results of Salt: https://tinyurl.com/5xbpt49u.)**
>
>
> - Both Experiment 2 and Experiment 3 demonstrate that our method can be extended to standard FWI datasets.
>
>
> [4] Multiscale Data-driven Seismic Full-waveform Inversion with Field Data Study, IEEE TGRS 2021
>
> [5] InversionNet: An efficient and accurate data-driven full waveform inversion, IEEE TGRS 2019
>
> [6] Deep-learning inversion: A next-generation seismic velocity model building method, Geophysics 2019
>
>
> **Typos/Misc**:
>
> - Thank you for point these out! The supervised method in the section 1 refers to H-PGNN+. We will revise our manuscript and use proper terms.

---

> > ### Comment · Reviewer_vGXF · 2021-11-27
> > **response**
> >
> > I'd like to thank the authors for the thorough response to the reviewer comments. The evaluation on other datasets is helpful to understand how robust the method is. While the method produces reasonable outputs for the other datasets, the performance is (expectedly) worse when generalizing to data that differ markedly from the training set.
> >
> > The authors should also be sure to clarify the differences to the related work in the paper. My main remaining criticism is that the novelty is perhaps somewhat limited compared to [1] and [2], which are also "unsupervised", but are not trained across a dataset.
> >
> > Still, if the additional evaluations and clarifications about related work can be included in the paper I would lean towards acceptance.

---

> > > ### Author Response · Authors · 2021-11-29
> > > **Response to Reviewer vGXF**
> > >
> > > **Q1: Still, if the additional evaluations and clarifications about related work can be included in the paper I would lean towards acceptance.**
> > >
> > > We promise that the additional evaluations and clarifications about related work will be included in the final draft. Specifically, our revision plan is as follows:
> > > - related work section will be rewritten as (see link: https://tinyurl.com/2p82shej).
> > > - evaluations on Marmousi/Salt, evaluations on more network architectures, and evaluations on noise handling will be added to the experiment section.
> > > - the rest of the experiments and all visualization will be added to the appendix.
> > >
> > > We are happy to provide the link to the new draft if required.
> > >
> > >
> > > **Q2: The authors should also be sure to clarify the differences to the related work in the paper. My main remaining criticism is that the novelty is perhaps somewhat limited compared to [1] and [2], which are also "unsupervised", but are not trained across a dataset.**
> > >
> > > Thanks for the suggestion. We further clarify the differences between our UPFWI and [1-3] as follows:
> > >
> > > **Difference to [1, 3]:** Compare to [1,3] (CNN-domain FWI), the novelty of our method is that it addresses the key limitation in these previous works — the initial guess, which is clearly discussed in [3] (Discussion page 12) as follows
> > >
> > >    > "CNN-domain FWI does not fundamentally address the need for good initial models in FWI."
> > >
> > >
> > > In [1, 3], the velocity map is generated by applying a network that is pretrained on an expert intial guess. The selection of the initial guess is crucial and costly. It is crucial as only carefully chosen guess can generate accurate prediction. It is costly as the careful choice of guess requires large effort from domain experts. The following link (https://tinyurl.com/2p9bwv49) shows the initial guess used in [1], which is blurry but clearly has similar pattern with the final result. Our method addresses this limitation by shifting the design paradigm to apply the network on the input seismic data (rather than on initial guess). Thus, the costly initial guess is not needed any more. Moreover, this paradigm shift introduces another benefit — learning a single mapping from seismic data to velocity map for multiple samples in a dataset or a sub-domain.
> > >
> > >
> > > In summary, previous works rely on carefully chosen initial guess and learn individual mapping per sample, while our method is initial guess free and learn a shared mapping for a dataset (or sub-domain).
> > >
> > >
> > > **Difference to [2]:** Before we compare to [2], let us clarify what *unsupervised* refers to in our paper. Here, *unsupervised* refers to the condition that *no* velocity map is available during either training or pre-training. In this sense, [2] is *not unsupervised* as it requires the ground truth of velocity maps to learn the generative model, which is used to search the optimal velocity map per sample during inference. By contrast, our UPFWI does not need any ground truth velocity maps.
> > >
> > >
> > > [1] Reparameterized full-waveform inversion using deep neural networks, Geophysics, 2021
> > >
> > > [2] Stochastic seismic waveform inversion using generative adversarial networks as a geological prior, Mathematical Geosciences, 2020
> > >
> > > [3] Parametric convolutional neural network-domain full-waveform inversion, Geophysics, 2019

---

> > > ### Author Response · Authors · 2021-12-07
> > > **Response to Reviewer vGXF [Follow Up]**
> > >
> > >
> > > Dear reviewer, we sincerely thank you for your valuable suggestions to help us improve our paper. Also, we would love to hear any feedback or comment from you on our response and provide additional information to answer your questions. Thank you very much.

---

> ### Author Response · Authors · 2021-11-23
> **Response to Reviewer vGXF [Part 1]**
>
> **Q1: Are the authors claiming the method as a contribution? The authors should clarify the novelty of their proposed method.**
>
> - Yes, we claim our proposed method as a contribution. Different from previous works either focusing on unsupervised learning per sample or supervised learning per dataset, our method achieves unsupervised learning per dataset. This results in different mathematical formulations and solutions. Below, we will explain the difference in details.
>
>
> - Our method is ***unsupervised*** learning per ***dataset***, formulated as:
> $$\boldsymbol{\hat{v}}(\boldsymbol{p}) = g_{\theta^{*}}(\boldsymbol{p});\mathrm{s.t.}~\theta^\*(\boldsymbol{\Phi}_u)=\underset{\theta}{\operatorname{argmin}} \underset{\boldsymbol{p}\in \boldsymbol{\Phi}_u }\sum \mathcal{L}(f(g_\theta(\boldsymbol{p_i})),\boldsymbol{p}_i),$$
> where  $\boldsymbol{\Phi}_u$ represents an unsupervised dataset that contains seismic data $\boldsymbol{p_i}$ alone, $\boldsymbol{p}$ is the seismic measurements, $\boldsymbol{v}$ is the velocity map. $\theta$ represents the trainable weights. $g_\theta(\cdot)$ is the network for inversion, $f(\cdot)$ is the forward modeling, and $\mathcal{L}(\cdot,\cdot)$ is a loss function.
>
>
> - In contrast, [1] and [2] perform ***unsupervised*** learning per *sample*,  formulated as:
> $$\boldsymbol{\hat{v}}(\boldsymbol{p}) = g_{\theta^\*(\boldsymbol{p})}(\boldsymbol{a})\; \mathrm{s.t.}~\theta^\*(\boldsymbol{p}) =\underset{\theta}{\operatorname{argmin}}\mathcal{L}(f(g_\theta(\boldsymbol{a})),\boldsymbol{p}),$$
> where $\boldsymbol{a}$ is a tensor with real numbers in the interval of $[-1, 1]$.
> Different network architectures have been proposed for $g_\theta(\cdot)$. Particularly, [1] develops a network based on convolution \& deconvolution layers, while [2] proposes one based on GAN.
>
>
> - In addition, [3] perform *supervised* learning per ***dataset***, formulated as
> $$\boldsymbol{\hat{v}}(\boldsymbol{p}) =  g_{\theta^*}(\boldsymbol{p})\;\mathrm{s.t.}~\theta^\*(\boldsymbol{\Phi_s})=\underset{\theta}{\operatorname{argmin}}\sum_{(\boldsymbol{v}_i, \boldsymbol{p}_i)\in\boldsymbol{\Phi}_s }\mathcal{L}(g_\theta(\boldsymbol{p_i}),\boldsymbol{v}_i) ,$$where $\boldsymbol{\Phi}_s$ represents a supervised dataset that includes not only seismic data $\boldsymbol{p_i}$ but also ground truth of velocity maps $\boldsymbol{v_i}$.
>
>
> - The difference between our proposed method and the work of [3] is straightforward (i.e. unsupervised vs. supervised).
>
>
> - Different from [1] and [2] where an individual model is optimized per sample, we learn a mapping function for a dataset rather than a sample. Our method has two advantages. First, in applications such as time-lapse imaging, where the variation over time for the same location is small, the initial learned model can be directly used without further training. This is validated in our experiment. https://tinyurl.com/56tatezx
>
> - Another advantage is that [1] and [2] require an *expert* initial guess which is crucial for optimization and expensive to acquire. As mentioned in [1], obtaining this initial guess is rather challenging. On the contrary, this can be saved in our UPFWI.
>
>
> [1] Reparameterized full-waveform inversion using deep neural networks, Geophysics 2021
>
> [2] Stochastic seismic waveform inversion using generative adversarial networks as a geological prior, Mathematical Geosciences 2020
>
> [3] Deep-learning inversion:  A next-generation seismic velocitymodel building method, Geophysics 2019

---

### Author Response · Authors · 2021-11-23
**Summary Response [Part 2]**

3. **Additional experiments on more challenging datasets (Marmousi \& Salt datasets).**
Our proposed model can be applied to more complex and realistic subsurface test datasets. We further evaluate UPFWI on two more challenging tests including Salt and Marmousi datasets and achieve solid results on both datasets. Click the following link for results:  https://tinyurl.com/5xbpt49u, https://tinyurl.com/5bbzr27p. The table below shows the quantitative results on both datasets. Although our UPFWI achieves good results on Salt dataset with preserved subsurface structures, it has clearly larger errors than the supervised InversionNet. This is due to two reasons: (a) Salt dataset has a small amount of training data (120 samples), which is very challenging for unsupervised methods; (b) the variability between training and testing samples is small, providing a significantly larger favor to supervised methods than the unsupervised counterparts.

    |Dataset | Model | MAE$\downarrow$ | MSE$\downarrow$ | SSIM$\uparrow$|
|---|---|---|---|---|
|Marmousi|InversionNet|149.67 | 45936.23 | 0.7889|
|Marmousi|UPFWI| 221.93 | 125825.75 | 0.7920|
|Salt|InversionNet | 25.98 | 8669.98 | 0.9764 |
|Salt| UPFWI | 150.34 | 164595.28 | 0.7837 |

    **(Visualization results of Salt: https://tinyurl.com/5xbpt49u.)**

    **(Visualization results of Marmousi: https://tinyurl.com/5bbzr27p.)**


4. **Additional experiments on more network architectures (Vision Transformer \& MLP-Mixer).** We further conducted experiments by using Vision Transformer (ViT) and MLP-Mixer to replace CNN as the encoder. See results in the following link: https://tinyurl.com/9bppamzs. The table below shows the quantitative results. Solid results are achieved for both network architectures, indicating our proposed method is model-agnostic.

    |Network Architecture | MAE$\downarrow$ | MSE$\downarrow$ | SSIM$\uparrow$ |
|---|---|---|---|
|CNN | 16.27 | 1705.35 | 0.9866 |
|ViT | 41.44 | 11029.01 | 0.9461 |
|MLP-Mixer | 22.32 | 4177.37 | 0.9726 |

    **(Visualization results of ViT and MLP-Mixer: https://tinyurl.com/9bppamzs.)**


5. **Additional experiments to investigate generalization.** We conducted two additional experiments: (1) training our model on the CurvedFault dataset and further testing on the FlatFault dataset (see results: https://tinyurl.com/thy2ak9e, quantitative results are shown in the table below) (2) testing our model on time-lapse imaging problems (see results: https://tinyurl.com/56tatezx). The results demonstrate that our proposed model yields generalization ability to a certain degree.

    |Training Dataset | Test Dataset| MAE$\downarrow$ | MSE$\downarrow$ | SSIM$\uparrow$ |
|---|---|---|---|---|
|FlatFault |FlatFault| 14.60 | 1146.09 | 0.9895 |
|CurvedFault |FlatFault|50.80 | 17627.65 | 0.9253 |

    **(Visualization results of generalization test on FlatFault: https://tinyurl.com/thy2ak9e.)**

    **(Visualization results of time-lapse imaging: https://tinyurl.com/56tatezx.)**


6. **Additional experiments on noise handling.** We validate the robustness of our UPFWI models by two additional tests: (1) testing data contaminated by Gaussian noise (see results: https://tinyurl.com/vn8fhenc, quantitative results are shown in Table 1) and (2) testing data with some missing traces (see results: https://tinyurl.com/344xnbas, quantitative results are shown in Table 2). The results demonstrate that our model can be robust to a certain levels of noise \& irregular acquisition.

    Table 1. Quantitative results of our UPFWI tested on seismic inputs with different noise levels.

    |Dataset | $\sigma$ ($10^{-4}$) | PSNR | MAE$\downarrow$ | MSE$\downarrow$ | SSIM$\uparrow$|
|---|---|---|---|---|---|
|FlatFault| 0 | 100 | 14.60 | 1146.09 | 0.9895|
|FlatFault| 0.5 | 61.60 | 15.68 | 1343.21 | 0.9888|
|FlatFault| 1.0 | 58.70 | 24.84 | 4010.78 | 0.9733|
|FlatFault |5.0 | 51.58 | 44.33 | 7592.57 | 0.9681|
| | | | | | |
|CurvedFault| 0 | 100 | 23.56 | 3639.96 | 0.9756|
|CurvedFault|0.5 | 61.72 | 23.78 | 3704.00 | 0.9751|
|CurvedFault|1.0 | 58.70 | 24.84 | 4010.78 | 0.9733|
|CurvedFault |5.0 | 51.68 | 46.90 | 10415.38 | 0.9441|

    **(Visualization results of adding Gaussian noise:  https://tinyurl.com/vn8fhenc.)**




    Table 2. Quantitative results of our UPFWI tested on seismic inputs with different numbers of missing traces.

    |Dataset | Missing Traces | MAE$\downarrow$ | MSE$\downarrow$ | SSIM$\uparrow$|
|---|---|---|---|---|
|FlatFault | 0 | 14.60 | 1146.09 | 0.9895|
|FlatFault|4 (5\%) | 21.23 | 1772.05 | 0.9868|
|FlatFault| 7 (10\%) | 33.66 | 3504.25 | 0.9814 |
|FlatFault|17 (25\%) | 85.21 | 16731.69 | 0.9457|
| | | | | | |
|CurvedFault| 0 | 23.56 | 3639.96 | 0.9756|
|CurvedFault|4 (5\%) | 41.33 | 6914.12 | 0.9622|
|CurvedFault|7 (10\%) | 61.72 | 12445.90 | 0.9453|
| CurvedFault |17 (25\%) | 121.06 | 36770.77 | 0.8853|

    **(Visualization results of missing traces: https://tinyurl.com/344xnbas.)**

---

### Author Response · Authors · 2021-11-23
**Summary Response [Part 1]**

We thank the reviewers for their valuable comments, which we have used to revise and improve our manuscript. Those discussions and additional tests will be added to our draft upon revision. The detailed responses to each reviewer are posted individually.

1. **Clarification of *unsupervised*.**
In our paper, *unsupervised* learning of FWI refers to the methods that do not use velocity maps as supervision in either training or pre-training.

2. **Clarification of novelty.**
Different from previous works either focusing on unsupervised learning per sample, weakly-supervised learning per sample, or supervised learning per subdomain, our method achieves unsupervised learning per subdomain. Here, a subdomain refers to a collection of similar subsurface structures (e.g. FlatFault, CurvedFault, Salt, etc.). Below we explain the differences mathematically.
Let us denote $\boldsymbol{p}$ as the seismic measurements, $\boldsymbol{v}$ as the velocity map. $\theta$ represents the trainable weights. $g_\theta(\cdot)$ is the network for inversion, $f(\cdot)$ is the forward modeling, and $\mathcal{L}(\cdot,\cdot)$ is a loss function (such as mean absolute error).
    1. ***Unsupervised*** learning per ***subdomain*** (our UPFWI)
$$\boldsymbol{\hat{v}}(\boldsymbol{p}) = g_{\theta^{*}}(\boldsymbol{p});\mathrm{s.t.}~\theta^\*(\boldsymbol{\Phi}_u)=\underset{\theta}{\operatorname{argmin}} \underset{\boldsymbol{p}\in \boldsymbol{\Phi}_u }\sum \mathcal{L}(f(g_\theta(\boldsymbol{p_i})),\boldsymbol{p}_i),~~~~~~~~~~~~~~~~~~~~(a)$$where  $\boldsymbol{\Phi}_u$ represents an unsupervised dataset that contains seismic data $\boldsymbol{p_i}$ alone.
    2. ***Unsupervised*** learning per *sample* [1]
$$\boldsymbol{\hat{v}}(\boldsymbol{p}) = g_{\theta^\*(\boldsymbol{p})}(\boldsymbol{a})\; \mathrm{s.t.}~\theta^\*(\boldsymbol{p}) =\underset{\theta}{\operatorname{argmin}}\mathcal{L}(f(g_\theta(\boldsymbol{a})),\boldsymbol{p}),~~~~~~~~~~~~~~~~~~~~~~~~~~~(b)$$
where $\boldsymbol{a}$ is a tensor with real numbers in the interval of $[-1, 1]$.
    3. *Weakly-supervised* learning per *sample* [2]
$$\boldsymbol{\hat{v}}(\boldsymbol{z}^\*) = g_{\theta^\*}(\boldsymbol{z}^\*)\;\mathrm{s.t.}~\boldsymbol{z}^\*(\boldsymbol{p}) =\underset{\boldsymbol{z}}{\operatorname{argmin}}\mathcal{L}(f(g_{\theta^\*}(\boldsymbol{z})), \boldsymbol{p}),\theta^\*(\boldsymbol{\Phi_{v}})=\underset{\theta}{\operatorname{argmin}}\sum_{v_{i}\in \Phi_{v}}\mathcal{L_{\mathrm{GAN}}}(g_\theta(\boldsymbol{\alpha_{i}}), \boldsymbol{v_{i}}),~~~~~~~(c)$$where $\boldsymbol{\Phi_v}$ is a training dataset including numerous velocity maps. $\boldsymbol{z}$ and $\boldsymbol{\alpha_i}$ are tensors sampled from the normal distribution.
    4. *Supervised* learning per ***subdomain***
$$\boldsymbol{\hat{v}}(\boldsymbol{p}) =  g_{\theta^*}(\boldsymbol{p})\;
        \mathrm{s.t.}~\theta^\*(\boldsymbol{\Phi_s})=\underset{\theta}{\operatorname{argmin}}\sum_{(\boldsymbol{v}_i, \boldsymbol{p}_i)\in\boldsymbol{\Phi}_s }\mathcal{L}(g_\theta(\boldsymbol{p_i}),\boldsymbol{v}_i) ,~~~~~~~~~~~~~~~~~~~~~~~~~~~~~~~~~~~(d)$$where $\boldsymbol{\Phi}_s$ represents a supervised dataset that includes not only seismic data $\boldsymbol{p_i}$ but also ground truth of velocity maps $\boldsymbol{v_i}$.

    Both Eqs. (a) and (b) are unsupervised approaches, which do not require labeled training pairs. In comparison, Eqs. (c) and (d) are supervised method.

    There are two major distinctions between ours in Eq.(a) and the one in Eq.(b). Firstly, the optimal weights $\theta^*(\boldsymbol{\Phi}_u)$ in Eq.(a) is obtained based on the dataset $\boldsymbol{\Phi}_u$ for a subdomain, whereas the optimal $\theta^*(\boldsymbol{p})$ in Eq.(b) is optimized per sample $\boldsymbol{p}$. After training with sufficient seismic data, Eq.(a) yields a mapping from seismic data to velocity maps that can be directly used for inference *without* any optimization. In contrast, the optimal weights in Eq.(b) are sample dependent, requiring optimization for any new sample before inference.

    Another important difference is that Eq.(b) requires an *expert* initial guess which is crucial for optimization and expensive to acquire. As mentioned in [1], obtaining this initial guess is rather challenging. On the contrary, this can be saved in our UPFWI. As UPFWI learns a mapping for a subdomain which enables significant cost saving for applications such as time-lapse imaging where the variation over time for a location is small, the initial learned model can be directly used without further training. This is validated in our experiment. https://tinyurl.com/56tatezx

[1] Reparameterized full-waveform inversion using deep neural networks, Geophysics, 2021

[2] Stochastic seismic waveform inversion using generative adversarial networks as a geological prior, Mathematical Geosciences, 2020

---

> ### Author Response · Authors · 2021-11-29
> **Additional Clarification of Weakly-supervised Learning**
>
> - We want to further clarify what *weakly-supervised* refers to in our comment. Compared to *fully-supervised* learning methods that use matched pairs of seismic data and velocity maps for training, the *weakly-supervised* learning methods use velocity maps alone.

---

### Decision · Program_Chairs · 2022-01-20

**Decision:**

Accept (Poster)

**Comment:**

The paper presents an unsupervised method for learning Full-Waveform Inversion in geophysics, by combining a differentiable physics simulation with a CNN based inversion network.

The reviewers agreed that the paper was well written and described an important advance but were concerned about limited novelty and a potential sim2real gap. The authors responded to their critique with significant new experiments and clarified the novelty of their method relative to prior work.

Based on the author responses, I recommend acceptance.